# CLIP-OGD: An Experimental Design for Adaptive Neyman Allocation in Sequential Experiments

**Jessica Dai**
UC Berkeley
jessicadai@berkeley.edu

**Paula Gradu**
UC Berkeley
pgradu@berkeley.edu

**Christopher Harshaw**\*
MIT
charshaw@mit.edu

## Abstract

From clinical development of cancer therapies to investigations into partisan bias, adaptive sequential designs have become increasingly popular method for causal inference, as they offer the possibility of improved precision over their non-adaptive counterparts. However, even in simple settings (e.g. two treatments) the extent to which adaptive designs can improve precision is not sufficiently well understood. In this work, we study the problem of Adaptive Neyman Allocation in a design-based potential outcomes framework, where the experimenter seeks to construct an adaptive design which is nearly as efficient as the optimal (but infeasible) non-adaptive Neyman design, which has access to all potential outcomes. Motivated by connections to online optimization, we propose Neyman Ratio and Neyman Regret as two (equivalent) performance measures of adaptive designs for this problem. We present CLIP-OGD, an adaptive design which achieves $\widetilde{\mathcal{O}}(\sqrt{T})$ expected Neyman regret and thereby recovers the optimal Neyman variance in large samples. Finally, we construct a conservative variance estimator which facilitates the development of asymptotically valid confidence intervals. To complement our theoretical results, we conduct simulations using data from a microeconomic experiment.

## 1 Introduction

From medicine and public health to economics and public policy, randomized control trials are used in a variety of disciplines to investigate causal effects. Typically, treatment is assigned in a non-adaptive manner, where assignments are determined before any outcomes are observed. A sequential experimental approach, which adaptively assigns treatment based on previously observed outcomes, offers the possibility of more precise or high powered estimates of relevant causal effects. Adaptive experiments are run to develop clinical therapies for breast cancer [Barker et al., 2009], evaluate incentives to reduce partisan bias [Offer-Westort et al., 2021], and evaluate customer acquisition via online advertising [Schwartz et al., 2017], to name a few.

In this paper, we study the problem of Adaptive Neyman Allocation, which we informally define as follows. An optimal non-adaptive experimental design which minimizes variance of an estimator will depend on the unknown potential outcomes, rendering it infeasible to run. However, by adaptively choosing treatment assignments in a sequential manner based on observed outcomes, we can hope to guarantee that the variance of the estimator under the adaptive design converges to the optimal non-adaptive variance. The problem of Adaptive Neyman Allocation is to construct such an adaptive design which guarantees the variance converges to the (infeasible) optimal non-adaptive design.

---

\*Corresponding author

37th Conference on Neural Information Processing Systems (NeurIPS 2023).

An experimental design which sufficiently addresses the Adaptive Neyman Allocation problem offers the advantage of higher statistical power, relative to a broad class of fixed experimental designs. Practically speaking, this means that either smaller confidence intervals are obtained for a given number of experimental units, or that fewer units are required to achieve confidence intervals of a given length. In practice, this means that investigating causal effects can be cheaper—in terms of time, money, and other valuable resources—when adaptive experiments are run. Although several experimental designs have been proposed for this purpose [Hahn et al., 2011, Blackwell et al., 2022], none have provided formal guarantees that the optimal non-adaptive variance can be achieved and the effectiveness of such designs has recently been called into question Cai and Rafi [2022].

The main contributions of this work are as follows:

1. **Neyman Ratio and Regret**: We propose two (equivalent) performance measures of experimental designs for the problem of Adaptive Neyman Allocation: Neyman Ratio and Neyman Regret. We show that guarantees on the rates of these performance measures directly translate to guarantees on the convergence of variance to the Neyman variance.

2. **CLIP-OGD**: We propose the adaptive design CLIP-OGD, a variant of online stochastic projected gradient descent for which the Neyman regret is $\widetilde{\mathcal{O}}(\sqrt{T})$. This guarantees that the variance of the sequential effect estimator approaches the Neyman variance.

3. **Confidence Intervals**: By constructing a conservative variance estimator, we provide confidence intervals which guarantee asymptotic coverage of the average treatment effect.

In Section 7, we support these theoretical results with simulations using data from a microeconomic experiment. Our results rely on viewing the Adaptive Neyman Allocation problem through the lens of online convex optimization. However, as discussed in Section 4.2, due to the subtleties arising in the problem, we do not know of an existing online algorithm which directly obtains these results.

## 1.1 Related Work

We work within the potential outcomes framework for causal inference Neyman [1923], Rubin [1980], Imbens and Rubin [2015]. The idea of optimal treatment allocation dates back to Neyman [1934], where he demonstrates that sampling from treatments proportional to the within-treatment outcome variance will minimize the variance of standard estimators. Unfortunately, this type of design is not practically feasible when little is known about the statistics of outcomes from each treatment. Robbins [1952] highlights adaptive sampling as one of the more pressing open statistical problems at the time. In Chapter 5, Solomon and Zacks [1970] presents a survey of adaptive designs for survey sampling, but from a Bayesian perspective. More recently, Hahn et al. [2011] proposed a two stage design in a super-population setting, where data is uniformly collected from both arms in the first stage, statistics of the treatment arm are estimated, and a fixed probability derived from estimated statistics is used in the second stage. They derive the limiting distribution of the effect estimator under the two-stage design, which has a variance that is similar to, but asymptotically bounded away from the optimal Neyman variance. In a design-based setting, Blackwell et al. [2022] propose a similar two-stage approach and, through simulations, provide practical guidance on how to choose the length of the first stage. Although both of these works are motivated by achieving the Neyman variance, neither formally show that this is possible under the two-stage design.

While the goal in this paper is to increase the precision of treatment effect estimates, a variety of response-adaptive designs have been developed for various objectives, including reducing mean total sample size [Hayre and Turnbull, 1981] and reduction of harm reduction in null hypothesis testing [Rosenberger et al., 2001]. Eisele [1994] proposes the Doubly Adaptive Coin Based Design, which is a meta-algorithm for targeting various allocation proportions when outcomes are drawn i.i.d. from an exponential family. Hu and Rosenberger [2003] critiques many response-adaptive designs as being "mypoic strategies" which have "adverse effects on power", providing an asymptotic framework by which to judge adaptive design when the outcomes are i.i.d. and binary. This asymptotic evaluation framework was extended to continuous outcomes by Zhang and Rosenberger [2006]. An additional line of work has developed adaptive Bayesian methods for subgroup identification [Xu et al., 2014].

Causal inference under adaptively collected data has seen a variety of recent developments which are adjacent to, but distinct from, the problem of Adaptive Neyman Allocation. One line of research has been to construct estimators via re-weighting which ensure consistency and normality when data

is collected via bandit algorithms [Hadad et al., 2021, Zhang et al., 2020, 2021]. A second line of research has been to provide inferential methods which are valid under data-dependent stopping times [Wald, 1945, Howard et al., 2021, Ham et al., 2022]. Finally, Offer-Westort et al. [2021] propose an adaptive experimental design for improved selective inference, when only the effect of the best performing treatment is to be inferred.

## 2  Preliminaries

The sequential experiment takes place over $T$ rounds, where we assume that $T$ is fixed and known to the experimenter. At each iteration $t \in [T]$, a new experimental unit (e.g. clinical participant), enters into the experiment, so that there are $T$ units in total. In an abuse of notation, we identify units with their respective round $t \in [T]$. The experimenter assigns a (random) treatment $Z_t \in \{0, 1\}$ (e.g. drug or placebo) to the experimental unit. The unit has two real-valued potential outcomes $y_t(1), y_t(0)$ which are unknown to the experimenter and represent the unit's measured response to the treatment assignments (e.g. measured heart rate). The term "potential" is used here because while only one treatment is assigned and thus only one outcome is observed, both outcomes have the potential to be observed. At the end of the round, the experimenter sees the observed outcome $Y_t = \mathbf{1}[Z_t = 1]y_t(1) + \mathbf{1}[Z_t = 0]y_t(0)$.

### 2.1  Potential Outcomes Framework

In this paper, we adopt a *design-based framework* where the sequence of potential outcomes $\{y_t(1), y_t(0)\}_{t=1}^T$ is deterministic and the only source of randomness is treatment assignment it-self. In particular, we place no assumption on the homogeneity of the outcomes: they are not necessarily related to each other in any systematic way. Although the potential outcomes are deterministic, we introduce finite population analogues of various statistics. Define the finite population second moments $S(1)$ and $S(0)$ and correlation of the treatment and control outcomes $\rho$ to be

$$S(1)^2 = \frac{1}{T}\sum_{t=1}^T y_t(1)^2 \ , \quad S(0)^2 = \frac{1}{T}\sum_{t=1}^T y_t(0)^2 \ , \quad \text{and} \quad \rho = \frac{\frac{1}{T}\sum_{t=1}^T y_t(1)y_t(0)}{S(1)S(0)} \ .$$

Observe that the correlation between treatment and control outcomes is bounded $\rho \in [-1, 1]$. Although we refer to $\rho$ as the correlation, it also known as the cosine similarity and is generally not equal to the Pearson correlation coefficient. We remark that although the potential outcomes $y_t(1)$ and $y_t(0)$ are deterministic, the observed outcome $Y_t$ is random, as it depends on random treatment assignment. The natural filtration according to these rounds is denoted as $\mathcal{F}_1 \dots \mathcal{F}_T$, so that $\mathcal{F}_t$ captures all randomness before the sampling of $Z_t$, i.e. the treatments assigned and outcomes observed in previous rounds.

In this sequential setting, the mechanism for random treatment assignment can incorporate observed outcomes from previous experimental rounds. This treatment mechanism, referred to as the *experimental design*, is selected by and thus known to the experimenter. Formally, the experimental design is a sequence of functions $\{\Pi_t\}_{t=1}^T$ with signature $\Pi_t : (\{0, 1\} \times \mathbb{R})^{t-1} \to [0, 1]$ such that treatment is assigned as $\Pr(Z_t = 1 \mid \mathcal{F}_t) = \Pi_t(Z_1, Y_1, \dots Z_{t-1}, Y_{t-1})$. We denote $P_t = \Pr(Z_t = 1 \mid \mathcal{F}_t)$ as the (random) probability of treatment assignment at iteration $t$, given previously observed treatment assignments and outcomes.

The causal estimand of interest is the *average treatment effect*, defined as

$$\tau = \frac{1}{T}\sum_{t=1}^T y_t(1) - y_t(0) \ .$$

The average treatment effect captures the average counterfactual contrast between a unit's outcomes under the two treatment assignments. For example, this could be the average contrast of a clinical participant's heart rate under the drug or placebo. Individual treatment effects are defined as $\tau_t = y_t(1) - y_t(0)$, but they cannot be estimated without strong additional assumptions, as only one outcome is observed.

A standard estimator of the average treatment effect is the Horvitz–Thompson estimator, which weights observed outcome by the probability of their observation [Narain, 1951, Horvitz and Thompson, 1952]. For adaptive designs, the standard Horvitz–Thompson estimator is infeasible because

the marginal probability of treatment assignment $\Pr(Z_t = 1)$ depends on the unknown potential outcomes. For this reason, we investigate the *adaptive Horvitz–Thompson estimator*, which uses the random (observed) treatment probabilities used at each iteration.

$$\hat{\tau} \triangleq \frac{1}{T} \sum_{t=1}^{T} Y_t \Big( \frac{\mathbf{1}[Z_t = 1]}{P_t} - \frac{\mathbf{1}[Z_t = 0]}{1 - P_t} \Big) \ ,$$

where we recall that $P_t = \Pi_t(Z_1, Y_1, \ldots Z_{t-1}, Y_{t-1})$ is the treatment probability under the experimental design given the observed data. When treatment assignments are non-adaptive and independent, then the adaptive Horvitz–Thompson estimator is the equivalent to the standard Horvitz–Thompson estimator. Such adaptively weighted estimators have been proposed previously in the literature, e.g. [Bowden and Trippa, 2015, Hadad et al., 2021]. Below, we provide positivity conditions under which the adaptive estimator is unbiased, and derive its variance.

**Proposition 2.1.** *If* $\min\{P_t, 1 - P_t\} > 0$ *almost surely for all* $t \in [T]$ *then the adaptive Horvitz–Thompson estimator is unbiased:* $\mathbb{E}[\hat{\tau}] = \tau$.

**Proposition 2.2.** *The variance of the adaptive Horvitz–Thompson estimator is*

$$T \cdot \mathrm{Var}(\hat{\tau}) = \frac{1}{T} \sum_{t=1}^{T} \Big( y_t(1)^2 \, \mathbb{E}\Big[\frac{1}{P_t}\Big] + y_t(0)^2 \, \mathbb{E}\Big[\frac{1}{1 - P_t}\Big] \Big) - \frac{1}{T} \sum_{t=1}^{T} \tau_t^2 \ .$$

## 2.2 Asymptotic Framework and Assumptions

Following the convention of design-based inference, we analyze statistical methods within an asymptotic framework [see e.g., Freedman, 2008, Lin, 2013, Sävje et al., 2021]. This provides a formal basis for reasoning about the performance of statistical methods as the sample size increases, giving meaning to conventional notions such as consistency and limiting distribution. Formally speaking, the asymptotic sequence of potential outcomes is a triangular array $\{\{y_{t,T}(1), y_{t,T}(0)\}_{t=1}^{T}\}_{T=1}^{\infty}$, which yields a sequence of estimands $\{\tau_T\}_{T=1}^{\infty}$ and, together with an appropriately specified sequence of experimental design, a sequence of estimators $\{\hat{\tau}_T\}_{T=1}^{\infty}$. Analysis which applies to a fixed $T$ is said to be finite-sample (e.g. $\mathbb{E}[\hat{\tau}_T] = \tau_T$) whereas analysis which applies to the entire sequence is said to be asymptotic (e.g. $\tau_T - \hat{\tau}_T \xrightarrow{p} 0$). Although we use an asymptotic framework, we emphasize that the majority of our results are derived from finite-sample analysis and are merely interpreted through the lens of the asymptotic framework. We drop the subscript $T$ for notational clarity.

The main regularity conditions we place on the sequence of potential outcomes is below.

**Assumption 1.** *There exist constants* $0 < c \leqslant C$ *with* $c < 1$ *such that for all* $T$ *in the sequence:*

1. **Bounded Moments**: $c \leqslant \big(\frac{1}{T} \sum_{t=1}^{T} y_t(k)^2\big)^{1/2} \leqslant \big(\frac{1}{T} \sum_{t=1}^{T} y_t(k)^4\big)^{1/4} \leqslant C \ \forall \ k \in \{0, 1\}$.

2. **Bounded Correlation**: $\rho \geqslant -(1 - c)$.

The upper moment bound in Assumption 1 stipulates that the potential outcomes cannot grow too large with the sample size, while the lower moment bound is a type of non-degeneracy condition that prevents an increasingly large fraction of the outcomes going to zero. These assumptions are analogous to finite fourth moment and positive second moment assumptions in an i.i.d. setting. The bounded correlation assumption stipulates that the treatment and control outcomes are not exactly negatively correlated. In this paper, we do not assume that these constants $C$ and $c$ are known to the experimenter; however, if the experimenter can correctly specify such bounds (perhaps knowing a priori the scaling of the outcomes) then some of the constant factors in our analysis can be improved. We emphasize here that Assumption 1 places no assumption on the order in which units arrive in the experiment. In this sense, Assumption 1 allows for arbitrary "non-stationarity" or "drift" in the potential outcomes over the experimental rounds. In the next section, these regularity assumptions will ensure that the Neyman variance converges to zero at the parametric rate.

## 3 Neyman Design: The Infeasible Non-Adaptive Ideal

The problem of Adaptive Neyman Allocation is to construct an adaptive experimental design that achieves nearly the same variance as an optimal non-adaptive experimental design, chosen with

knowledge of all potential outcomes. The optimal non-adaptive design, referred to as the Neyman Design, is infeasible to implement because it depends on all potential outcomes, which are unknown to the experimenter at the design stage. The goal is that an adaptive experimental design—which can select treatment assignment based on observed outcomes—can gather enough information to perform as well as the infeasible Neyman design.

In order to define the optimal non-adaptive design, we begin by defining the class of Bernoulli designs. Informally, the class of Bernoulli designs consists of non-adaptive designs where each unit receives treatment $Z_t = 1$ with probability $p$, independently of past treatment assignments and observations. Formally, this class is parameterized by a non-adaptive sampling probability $p \in [0, 1]$ such that for all $t \in [T]$, the treatment policy $\Pi_t$ is a constant function whose value is $p$. Using Proposition 2.2, we can derive the variance of the Bernoulli design with parameter $p \in [0, 1]$ to be

$$T \cdot V_p = S(1)^2 \Big(\frac{1}{p} - 1\Big) + S(0)^2 \Big(\frac{1}{1 - p} - 1\Big) + 2\rho S(1)S(0) \ .$$

From the above, we can see that in order to minimize the variance of the Horvitz–Thompson estimator under the Bernoulli design, we should set the sampling probability $p$ so as to balance the square of the second moments of treatment and control outcomes. The Neyman Design is the Bernoulli design which minimizes the variance of the Horvitz–Thompson estimator. The corresponding optimal probability $p^*$ and variance $V_N$ are referred to as the Neyman probability and Neyman variance, respectively. The following proposition derives these quantities in terms of the potential outcomes.

**Proposition 3.1.** *The Neyman variance is $T \cdot V_N = 2(1 + \rho)S(1)S(0)$, which is achieved by the Neyman probability $p^* = (1 + S(0)/S(1))^{-1}$.*

In order to quantify the reduction in variance achieved by the Neyman design, define the *relative Neyman efficiency with respect to $p \in [0, 1]$* to be $V_N/V_p$. Intuitively, this ratio is a scale-free measure which captures the percent reduction in variance of the sequential Horvitz–Thompson estimator under the Neyman design. Formally, the equation for the relative Neyman efficiency is given below:

$$\frac{V_N}{V_p} = 2(1 + \rho)\Bigg[\frac{S(1)}{S(0)} \cdot \frac{(1 - p)}{p} + \frac{S(0)}{S(1)} \cdot \frac{p}{(1 - p)} + \rho\Bigg]^{-1} \ .$$

Consider the setting where outcomes are uncorrelated, and treatment outcomes are larger than control outcomes, e.g. $\rho = 0$, $S(1) = 4 \cdot S(0)$. In this case, the Neyman design is able to achieve less than half the variance of the uniform Bernoulli design (with $p = 1/2$): we can plug in to 3 to see that in this setting, we have $V_N/V_p = 0.47$. The improvement is larger if the experimenter makes erroneous assumptions about the relative magnitudes of the treatment and control outcomes and attempts to set $p$ accordingly: for example, if the experimenter had set $p = 1/4$, incorrectly believing that $S(1) \leqslant S(0)$, then the Neyman allocation results in a sixfold improvement in variance. Blackwell et al. [2022] derives qualitatively similar analysis of Neyman efficiency for stratified designs.

While the relative Neyman efficiency is helpful in determining the variance reduction afforded by the (infeasible) optimal Bernoulli design, it does not address the main question: which adaptive experimental designs can guarantee similar variance reduction? In the next section, we propose a performance metric which better addresses this question.

## 4 Adaptive Neyman Allocation: An Online Optimization Approach

### 4.1 Neyman Ratio and Neyman Regret: New Performance Measures

Let $V$ be the variance of the adaptive experimental design. We introduce our first performance measure of a sequential experimental design for Adaptive Neyman Allocation.

**Definition 1.** The *Neyman ratio* of a sequential experimental design is $\kappa_T = (V - V_N)/V_N$.

The subscript $T$ in $\kappa_T$ in included the reflect dependence of the number of rounds $T$. The Neyman ratio is motivated by the following relationship between the adaptive variance and the optimal Neyman variance:

$$V = \Big(\frac{V}{V_N}\Big) \cdot V_N = \Big(1 + \kappa_T\Big) \cdot V_N \ . \tag{1}$$

Equation (1) shows that the adaptive design can recover the Neyman variance if and only if the Neyman ratio $\kappa_T$ can be made arbitrarily small. For this reason, we propose the Neyman ratio as a performance measure of a sequential experimental design.

A natural question then becomes: how small can the Neyman ratio $\kappa_T$ be made as the number of rounds $T$ increases? To answer this question, we view the problem of minimizing the Neyman ratio through the lens of online optimization. To this end, we must re-express the variance of the sequential experimental design. For each round $t \in [T]$, define the cost function $f_t : [0,1] \to \mathbb{R}$ as $f_t(p) = y_t(1)^2/p + y_t(0)^2/(1-p)$. Observe that by Proposition 2.2, the variance is given by $T \cdot \mathrm{Var}(\hat{\tau}) = \mathbb{E}[\frac{1}{T} \sum_{t=1}^{T} f_t(P_t)]$. This reformulation of variance does not allow us to minimize variance directly, for the usual reason that the outcomes, and thus the cost functions $f_t$, are not fully observed. On the other hand, our goal is only to show that the variance of the adaptive design is comparable to the Neyman variance.

**Definition 2.** The *Neyman regret* of a sequential experimental design is

$$\mathcal{R}_T = \sum_{t=1}^{T} f_t(P_t) - \min_{p \in [0,1]} \sum_{t=1}^{T} f_t(p) \ .$$

Recall that $P_t$ is the random treatment probability at round $t$. The Neyman regret compares the accumulated costs $f_t(P_t)$ incurred by the adaptive design to the accumulated costs incurred by the optimal Bernoulli design which has access to all potential outcomes. The Neyman regret is random because the sequence $P_1, \ldots P_T$ is random. The following theorem connects the expected Neyman regret to the Neyman ratio.

**Theorem 4.1.** *Under Assumption 1, the Neyman ratio is within a constant factor of the $1/T$-scaled expected Neyman regret:* $\kappa_T = \Theta(\frac{1}{T} \mathbb{E}[\mathcal{R}_T])$.

Theorem 4.1 demonstrates that the Neyman ratio can be made small by minimizing the expected Neyman regret in an online fashion. In particular, any sublinear bound on the expected Neyman regret ensures that the Neyman ratio goes to zero so that, in large samples, the adaptive design achieves the variance reduction of the optimal Neyman design. Any adaptive design which aims to achieve Neyman variance must, to some extent, minimize expected Neyman regret.

Fortunately, online optimization is a well-studied area with a rich source of techniques from which we may draw inspiration. However, to the best of our knowledge, existing regret minimization algorithms are not well-suited to minimizing the Neyman regret. For example, the multi-arm bandit literature typically defines regret in terms of a finite number of actions that can be taken [Lattimore and Szepesvári, 2020] while Adaptive Neyman Allocation consists of a continuum of actions as $P_t \in [0,1]$. This means that algorithms like UCB [Auer et al., 2002a] and EXP3 [Auer et al., 2002b] are not appropriate for Adaptive Neyman Allocation. Our cost objectives $f_t$ and action space $[0,1]$ are both convex, so the problem of Adaptive Neyman Allocation is an instance of Online Convex Optimization (OCO) [Hazan, 2016]. Even so, the problem of minimizing Neyman regret is not immediately amenable to existing algorithms, which typically requires assumptions on the cost functions such as bounded gradients or known Lipschitz parameters. In this setting, the cost functions have gradients which blow up at the boundary and Lipschitz parameters cannot be guaranteed as they rely on the unknown heterogeneous potential outcomes. For these reasons, we must design a new algorithm specifically tailored to Adaptive Neyman Allocation.

### 4.2 CLIP-OGD: A Variant of Online Stochastic Projected Gradient Descent

We present CLIP-OGD, which aims to minimize the Neyman regret and thus recover the Neyman variance in large samples. The algorithm is based on the online stochastic projected gradient descent principle, but with a twist: the projection set continuously grows over the rounds. At each round $t$, a new treatment probability $P_t$ is chosen by updating the previous sampling probability $P_{t-1}$ in the negative (estimated) gradient direction of the previous cost, and then projecting to an interval $[\delta_t, 1 - \delta_t]$. Initially, this projection interval contains only the point $1/2$ and it grows as the rounds increase, allowing for larger amounts of exploitation in later rounds.

The gradient estimator $G_t$ is obtained in the following way: the gradient of $f_t$ at $P_t$ is given as $f'(P_t) = -\frac{y_t(1)^2}{P_t^2} + \frac{y_t(0)^2}{(1-P_t)^2}$ . Only one outcome is observed, so we used the adaptive Horvitz–

Thompson principle using the conditional probability $P_t$ to unbiasedly estimate the outcomes. CLIP-OGD is formally presented below as Algorithm 1, where the projection operator is defined as $\mathcal{P}_c(x) = \max\{c, \min\{x, 1 - c\}\}$.

---

**Algorithm 1:** CLIP-OGD

---

**Input:** Step size $\eta$ and decay parameter $\alpha$
Initialize $P_0 \leftarrow 1/2$ and $G_0 \leftarrow 0$
**for** $t = 1 \ldots T$ **do**
    Set projection parameter $\delta_t = (1/2) \cdot t^{-1/\alpha}$
    Compute new treatment probability $P_t \leftarrow \mathcal{P}_{\delta_t}(P_{t-1} - \eta \cdot G_{t-1})$
    Sample treatment assignment $Z_t$ as 1 with probability $P_t$ and 0 with probability $1 - P_t$
    Observe outcome $Y_t = \mathbf{1}[Z_t = 1]y_t(1) + \mathbf{1}[Z_t = 0]y_t(0)$
    Construct gradient estimator $G_t = Y_t^2 \left( -\frac{\mathbf{1}[Z_t=1]}{P_t^3} + \frac{\mathbf{1}[Z_t=0]}{(1-P_t)^3} \right)$
**end**

---

Unlike the two-stage design of [Hahn et al., 2011, Blackwell et al., 2022], CLIP-OGD does not feature explicit explore-exploit stages, but rather performs both of these simultaneously. The trade-off is implicitly controlled through parameters $\eta$ and $\alpha$: smaller values of $\eta$ limit the amount of that sampling probabilities can update and, likewise, larger values of $\alpha$ prevent extreme probabilities in earlier stages. Because the gradient of the cost functions are inversely proportional to the treatment probabilities, limiting the extremeness of the treatment probabilities in this way ensures that the gradient estimates do not increase at a fast rate. By appropriately setting input parameters, CLIP-OGD achieves $\tilde{\mathcal{O}}(\sqrt{T})$ expected Neyman regret, where the $\tilde{\mathcal{O}}(\cdot)$ notation hides sub-polynomial factors.

**Theorem 4.2.** *Under Assumption 1 the parameter values $\eta = \sqrt{1/T}$ and $\alpha = \sqrt{5\log(T)}$ ensure the expected Neyman regret of* CLIP-OGD *is asymptotically bounded: $\mathbb{E}[\mathcal{R}_T] \leqslant \tilde{\mathcal{O}}(\sqrt{T})$.*

Theorem 4.2 answers, in the affirmative, that it is possible to construct an adaptive experimental design whose variance recovers that of the Neyman variance, in large samples. Note that the amount of exploration (as given by the parameters $\eta$ and $\alpha$) should be increasing with $T$ in order to recover these regret bounds. In Appendix C, we show that CLIP-OGD is somewhat robust to different values of the decay parameter, i.e. for any value $\alpha > 5$, the expected regret will be sublinear. We also show that if the experimenter presumes to have correctly specified bounds $C$ and $c$ appearing in Assumption 1, then the step size can be modified to improve the constant factors in the Neyman regret bound, which may lead to improved performance in moderate sample sizes. We conjecture that the minimax rate for expected Neyman regret is $\mathcal{O}(\sqrt{T})$, but proving this is beyond the scope of the current paper—we only remark that we do not know it to immediately follow from any existing regret lower bounds for OCO.

## 5   Inference in Large Samples

The proposed CLIP-OGD was constructed to ensure that the variance of the adaptive Horvitz–Thompson estimator quickly approaches the Neyman variance. In this section, we provide confidence intervals for the average treatment effect which also enjoy reduced width compared to non-adaptive counterparts.

A necessary condition for variance estimation is that the variance itself cannot be going to zero too quickly. In design-based inference, it is common to directly posit a so-called "non-superefficient" assumption that $\text{Var}(\hat{\tau}) = \Omega(1/T)$ [Aronow and Samii, 2017, Leung, 2022, Harshaw et al., 2022]. The non-superefficiency assumption may be seen as an additional regularity assumption on the outcomes, e.g. preventing $y_t(1) = y_t(0) = 0$ for all $t \in [T]$. In this work, a similar lower bound on the rate of the adaptive variance is obtained through a different, perhaps more transparent, assumption on the expected Neyman regret.

**Assumption 2.** The outcome sequence is not overly-fit to CLIP-OGD: $-\mathbb{E}[\mathcal{R}_T] = o(T)$.

While we have shown that $\mathbb{E}[\mathcal{R}_T] \leqslant \tilde{\mathcal{O}}(\sqrt{T})$, the Neyman regret could in principle be negative if the adaptive design achieves variance which is strictly smaller than the best Bernoulli design. While this

seems unlikely to happen for "typical" outcomes, it is not impossible. Assumption 2 rules out these edge-case settings. We suspect that Assumption 2 would not be necessary in an i.i.d. setting, but proving this seems beyond the scope of the current paper. As shown in the appendix, Assumptions 1 and 2 imply that the adaptive variance achieves the parametric rate: $\text{Var}(\hat{\tau}) = \Theta(1/T)$.

## 5.1 Variance Estimation

In this section, we provide a variance estimator and show its stability in large samples. Rather than estimating the adaptive variance (which has no simple closed form), our approach is to estimate the Neyman variance directly. For an adaptive design achieving sublinear expected Neyman regret, these two quantities are asymptotically equivalent. In this way, our variance estimator may be appropriate not only for CLIP-OGD, but for any adaptive design achieving sublinear expected Neyman regret.

Recall that the Neyman variance is given by $T \cdot V_{\text{N}} = 2(1 + \rho)S_1 S_0$, where $\rho$ is the outcome correlation, $S_1$ is the second moment of treatment outcomes and $S_0$ is the second moment of control outcomes. Unfortunately, the outcome correlation term is generally not estimable without strong assumptions in a design-based framework. Indeed, the difficulty is that terms like $y_t(1)y_t(0)$ are unobservable due to the fundamental problem of causal inference [Imbens and Rubin, 2015]. A common solution to the problem is to opt for a conservative estimate of the variance, which will ensure validity of resulting confidence intervals.

We propose estimating the following upper bound on the variance: $T \cdot \text{VB} = 4S_0 S_1$. This upper bound on the Neyman variance is tight (i.e. $\text{VB} = V_{\text{N}}$) when outcome correlation satisfies $\rho = 1$. For example, this occurs when all individual treatment effects are zero, i.e. $y_t(1) = y_t(0)$ for all $t \in [T]$. Conversely, the upper bound will become looser for smaller values of the outcome correlation. In this sense, our bound resembles both the Neyman bound and the Aronow-Samii bound [Neyman, 1923, Aronow and Samii, 2013]. It may be possible to use the recent insights of Harshaw et al. [2021] in order to construct variance bounds which are tight in other scenarios, but that is beyond the scope of the current paper. Our variance estimator is defined as

$$T \cdot \widehat{\text{VB}} \triangleq 4\sqrt{\left(\frac{1}{T}\sum_{t=1}^{T} y_t^2 \frac{\mathbf{1}[z_t = 1]}{p_t}\right) \cdot \left(\frac{1}{T}\sum_{t=1}^{T} y_t^2 \frac{\mathbf{1}[z_t = 0]}{1 - p_t}\right)} \ ,$$

which is essentially a plug-in Horvitz-Thompson estimator for the second moments. Theorem 5.1 shows the error of the normalized variance estimator converges at a parametric rate.

**Theorem 5.1.** *Under Assumptions 1 and 2, and the parameters stated in Theorem 4.2, the error of the normalized variance estimator under* CLIP-OGD *is* $T \cdot \widehat{\text{VB}} - T \cdot \text{VB} = \widetilde{\mathcal{O}}_p(T^{-1/2})$.

## 5.2 Confidence Intervals

The variance estimator may be used to construct confidence intervals for the average treatment effect. This offers experimenters standard uncertainty quantification techniques when running CLIP-OGD. The following corollary shows that the resulting Chebyshev-type intervals are asymptotically valid.

**Corollary 5.1.** *Under Assumptions 1 and 2, and parameters stated in Theorem 4.2, Chebyshev-type intervals are asymptotically valid: for all* $\alpha \in (0, 1]$, $\liminf_{T \to \infty} \Pr(\tau \in \hat{\tau} \pm \alpha^{-1/2}\sqrt{\widehat{\text{VB}}}) \geqslant 1 - \alpha$.

While these confidence intervals are asymptotically valid under our regularity assumptions, they may be overly conservative in general. In particular, they will over cover when the Chebyshev tail bound is loose. We conjecture that the adaptive Horvitz–Thompson estimator under CLIP-OGD satisfies a Central Limit Theorem, which would imply asymptotic validity of the narrower Wald-type intervals where $\alpha^{-1/2}$ scaling is replaced with the corresponding normal quantile, $\Phi^{-1}(1 - \alpha/2)$. As discussed in Section 7, the adaptive estimator appears approximately normal in simulations. Until this is formally shown, we recommend experimenters exhibit caution when using Wald-type confidence intervals for the adaptive Horvitz–Thompson estimator under CLIP-OGD.

## 6 Considering Alternative Designs

**Explore-Then-Commit** Two-stage adaptive designs have been proposed for the purpose of variance reduction [Hahn et al., 2011, Blackwell et al., 2022]. Due to its similarities to algorithms in the bandits

literature, we call these types of designs Explore-Then-Commit (ETC) [Lattimore and Szepesvári, 2020]. At a high level, an Explore-then-Commit design runs the Bernoulli design with $p = 1/2$ for $T_0 \leqslant T$ iterations, uses the collected data to estimate $p^*$ by $\widehat{p}^*$, and then runs the Bernoulli design with $p = \widehat{p}^*$ for the remaining $T_1 = T - T_0$ iterations. These ETC designs are conceptually simpler than CLIP-OGD, and may be reasonable to apply in more restricted settings where changing the treatment probabilities is difficult or costly. However, we provide the following negative result which shows that they can suffer linear Neyman regret.

**Proposition 6.1.** *For all explore phase lengths $T_0$ satisfying $T_0 = \Omega(T^\epsilon)$ for some $\epsilon > 0$, there exist a class of potential outcomes sequences satisfying Assumption 1 such that the Neyman regret under Explore-then-Commit is linear: $\mathcal{R}_T = \Omega_p(T)$.*

The specific class of potential outcomes referenced in Propposition 6.1 is constructed explicitly in Appendix E.1. ETC designs suffer larger variance when the estimated $\widehat{p}^*$ may be far from the true optimal probability $p^*$. In a design-based setting, this happens when the units in the explore phase are not representative of the entire sequence. Formulating conditions under which Explore-then-Commit designs achieve low Neyman regret is beyond the scope of this paper, but the proof of Proposition 6.1 shows that additional regularity conditions on the order of the units will be required.

**Multi Arm Bandit Algorithms**    Multi Arm Bandit (MAB) algorithms are often used for adaptive decision making settings, from online advertising to product development. The goal of MAB algorithms is to minimize the outcome regret, which measures the contrast between the overall value obtained from the actions relative to the value of the best action. The outcome regret is conventionally defined as $\mathcal{R}_T^{\text{outcome}} = \max_{k \in \{0,1\}} \sum_{t=1}^T y_t(k) - \sum_{t=1}^T Y_t$. In certain contexts, minimizing outcome regret may be a more desirable goal than estimating a treatment effect to high precision. However, the following proposition illustrates that these two objectives are generally incompatible.

**Proposition 6.2.** *Let $\mathcal{A}$ be an adaptive treatment algorithm achieving sublinear outcome regret, i.e. there exists $q \in (0, 1)$ such that $\mathbb{E}[\mathcal{R}_T^{\text{outcome}}] \leqslant O(T^q)$ for all outcome sequences satisfying Assumption 1. Then, there exists a class of outcome sequences satisfying Assumption 1 on which $\mathcal{A}$ suffers super-linear Neyman regret, i.e. $\mathbb{E}[\mathcal{R}_T] \geqslant \Omega(T^{2-q})$.*

Proposition 6.2 demonstrates that the outcome regret and the Neyman regret cannot generally be simultaneously minimized. In particular, sublinear outcome regret implies that the variance of the estimator must converge slower than the $\Theta(1/T)$ parametric rate. This result contributes to a growing body of work which highlights trade-offs between various possible objectives in sequential decision making [Burtini et al., 2015]. It is beyond the scope of the current paper to determine how such trade-offs ought to be resolved, though Appendix F discusses ethical considerations.

# 7    Numerical Simulations

We evaluate the performance of CLIP-OGD and Explore-then-Commit (ETC) for the purpose of Adaptive Neyman Allocation on the field experiment of Groh and McKenzie [2016], which investigates the effect of macro-insurance on micro-enterprises in post-revolution Egypt[2]. The experimental units are 2,961 clients of Egypt's largest microfinance organization and the treatment was a novel insurance product. Several outcomes were recorded including whether the clients took on loans, introduced a new product or service, and the amount invested in machinery or equipment following treatment. To allocate treatment, Groh and McKenzie [2016] use a non-adaptive matched pair experimental design. Our goal here is not to provide a new analysis of this study, but rather to construct a plausible experimental setting under which to evaluate adaptive experimental designs.

In our simulations, we focus on the numerical outcome "invested in machinery or equipment". The experimental data contains only observed outcomes, so we must impute the missing potential outcomes in order to simulate the experiment. We impute outcomes using the model $y_t(1) - y_t(0) = \tau + \gamma_t$, where $\tau = 90,000$ and $\gamma_1 \ldots \gamma_T \sim \mathcal{N}(0, \sigma^2)$ are independent with $\sigma = 5,000$. This randomness is invoked only to impute potential outcomes, i.e. not re-sampled during each run of the experiment. In order to increase the sample size, we create a larger population by repeating this processes 5 times, which yields a total of $14,445$ units after those with missing entries are removed. Units are shuffled to appear in an arbitrary order and outcomes are normalized to be in the range $[0, 1]$.

---

[2]A repository for reproducing simulations is: `https://github.com/crharshaw/Clip-OGD-sims`

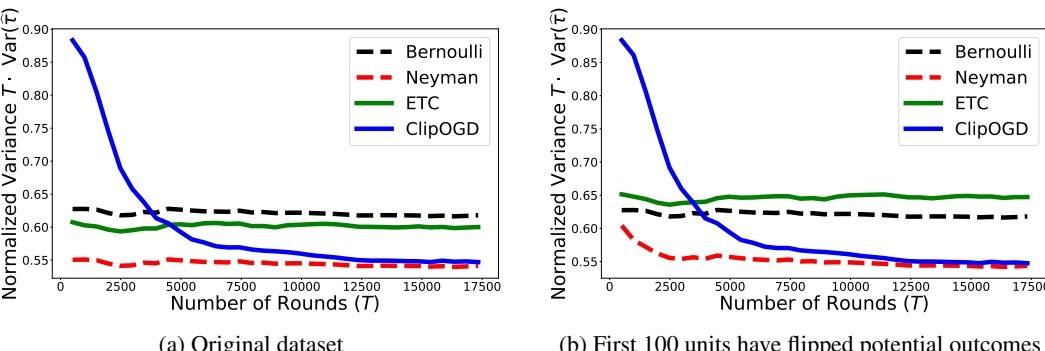

(a) Original dataset

(b) First 100 units have flipped potential outcomes

Figure 1: Normalized Variance of Adaptive Estimator under Experimental Designs

Figure 1 presents two plots illustrating how the variance of the adaptive HT estimator varies with different designs. The $x$ axis contains the number of rounds $T$ and the $y$ axis contains the normalized variance $T \cdot \mathrm{Var}(\hat{\tau})$ under the designs. For each value of $T$, we take the population to be the first $T$ units in the sequence. CLIP-OGD is run with the parameters recommended in Theorem 4.2 and ETC is run with $T_0 = T^{1/3}$ so that the exploration phase grows with $T$. The variance under CLIP-OGD and ETC is estimated empirically from 50,000 runs of the experiment, while the variance under the Bernoulli and Neyman designs is computed exactly.

In Figure 1a, we observe that CLIP-OGD requires about $T = 4,000$ samples to achieve variance equal to Bernoulli, but eventually converges to the Neyman variance. As discussed in Section 4.2, it may be possible to improve the convergence rate by incorporating knowledge of the outcome moments in the design parameters. On the other hand, ETC remains comparable with Bernoulli even for small values of $T$, but remains far away from the Neyman design for large samples. In Figure 1b, a similar simulation is run, except that the potential outcomes of the first 100 units are swapped, so that the first units have negative individual treatment effects. While this produces little effect on the performance of CLIP-OGD, it substantially worsens the performance of ETC, which relies on the early outcomes to estimate an optimal treatment probability. In particular, ETC performs worse than Bernoulli under this minor modification—even in large samples—corroborating Proposition 6.1.

In the appendix, we evaluate the proposed confidence intervals, showing that CLIP-OGD enjoys intervals of reduced width. We show that normal based intervals cover at the nominal level and provide further evidence that the estimator is asymptotically normal under CLIP-OGD. We run additional simulations to investigate the sensitivity of the step size, and to demonstrate that additional baselines which were not designed for Neyman allocation indeed perform poorly.

## 8  Conclusion

In this paper, we have proposed the Neyman ratio and Neyman regret as a performance measure of experimental designs for the Adaptive Neyman Allocation problem. To this end, we proposed CLIP-OGD which achieves $\widetilde{\mathcal{O}}(\sqrt{T})$ expected Neyman regret under mild regularity conditions on the outcomes. This formally establishes—for the first time—the existence of adaptive experimental designs under which the variance of the effect estimator quickly approaches the Neyman variance. Finally, we have provided a variance estimator which provides experimenters with uncertainty quantification methods when using CLIP-OGD. The main drawback of our analysis is that it is most relevant for moderate and large sample sizes; in particular, our work does not properly address whether adaptive designs are always beneficial in small samples.

There are several research directions which can improve relevance of this methodology to practice. First, establishing conditions under which a central limit theorem holds for CLIP-OGD will yield smaller and thus more desirable Wald-type confidence intervals. Second, investigations into batched treatment allocations and delayed observations of outcomes would allow practitioners more flexibility in their designs. Finally, investigating variants of Adaptive Neyman Allocation in the presence of interference [Aronow and Samii, 2017, Harshaw et al., 2022] would allow for more realistic inference in complex settings, e.g. social network experiments and marketplace experiments.

## Acknowledgments and Disclosure of Funding

We thank P.M. Aronow, Molly Offer-Westort, Alexander Rakhlin, Benjamin Recht, Fredrik Sävje, and Daniel Spielman for insightful discussions which helped shaped this work. Part of this work was done while Christopher Harshaw was visiting the Simons Institute for the Theory of Computing. Christopher Harshaw gratefully acknowledges support from Foundations of Data Science Institute (FODSI) NSF grant DMS2023505.

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

# Appendix

## Table of Contents

# A  Additional Simulation Results

In this section, we present additional simulation results on the Groh and McKenzie [2016] data. We refer to Section 7 for a review of the experimental set-up. In this section, we focus on the full dataset where $T = 14,445$. Simulations were run on a 2019 MacBook Pro with 2.4 GHz Quad-Core Intel Core i5 and 16 GB LPDDR3 RAM.

## A.1  Confidence Intervals

|  | Chebyshev Width | Chebyshev Coverage | Normal Width | Normal Coverage |
|---|---|---|---|---|
| Bernoulli ($p = 1/2$) | 0.0541 | 100% | 0.0237 | 95.21% |
| CLIP-OGD | 0.0507 | 99.99% | 0.0222 | 95.22% |

Table 1: 95% Confidence Intervals for Bernoulli and CLIP-OGD

Table 1 presents the Chebyshev-based and Normal-based intervals for the Bernoulli design ($p = 1/2$) and CLIP-OGD. We see that while Chebyshev over-covers, the normal-based confidence intervals cover at the nominal level with reduced width for both designs. The relative Neyman efficiency on this dataset is somewhat close to 1, so that the reduction of the width of the confidence intervals afforded by CLIP-OGD is present, though modest. The coverage of the normal-based confidence intervals provides further evidence supporting our conjecture that the adaptive Horvitz–Thompson estimator is asymptotically normal under CLIP-OGD.

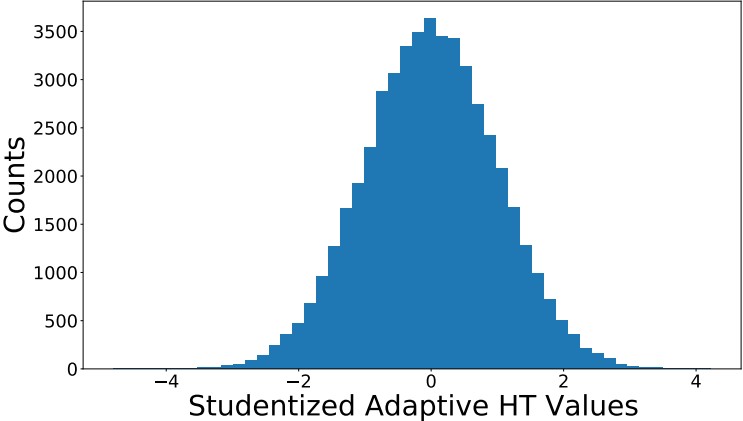

Figure 2: Histogram of Studentized Adaptive Horvitz–Thompson estimator under CLIP-OGD ($T = 14,445$)

Figure 2 plots the histogram of the studentized adaptive Horvitz–Thompson estimator under CLIP-OGD. By studentized, we mean that the histogram is plotting the draws of the random variable

$$Z = \frac{\tau - \hat{\tau}}{\sqrt{\mathrm{Var}(\hat{\tau})}} \quad .$$

We estimate the standard deviation empirically from 50,000 runs of the experiment. The estimator is said to be asymptotically normal if $Z \xrightarrow{d} \mathcal{N}(0,1)$. Figure 2 provides evidence that asymptotic normality is likely to hold in this setting. Formally establishing asymptotic normality is beyond the scope of the current paper as it would involve very different analytic techniques than those used to establish sublinear Neyman regret.

## A.2  Sensitivity to Step Size

In this section, we explore through simulations how the performance of CLIP-OGD depends on the step size.

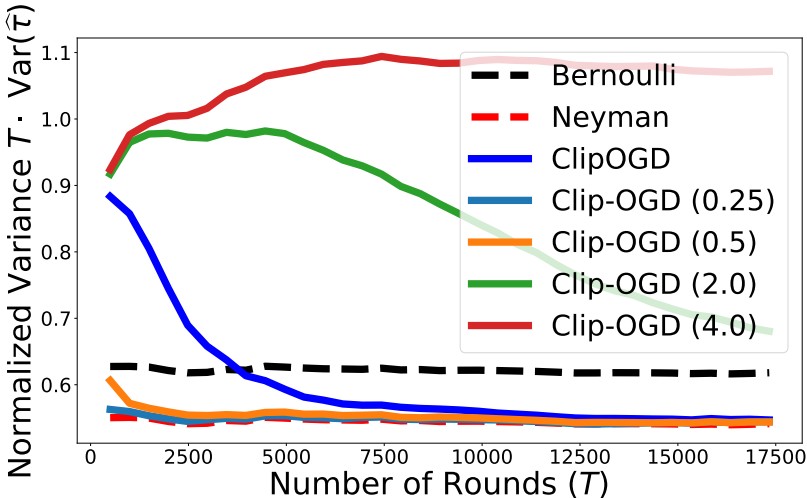

Figure 3: Comparing Step Sizes:

In Figure 3, we re-create Figure 1a in the main paper but we have added instances of CLIP-OGD with different step sizes of the forms $\eta = c/\sqrt{T}$ for $c \in \{0.25, 0.5, 1.0, 2.0, 4.0\}$. We find that smaller step sizes improve convergence rates, effectively removing the "overhead of adaptivity" in this example. However, because the randomized experiment can only be run once, experimenters will typically not be able to try many step sizes. While it remains an open question about how to select a step size which best mitigates the "overhead of adaptivity", our recommendation of $1/\sqrt{T}$ still maintains good convergence properties.

### A.3 Alternative Designs

In this section, we conduct additional experiments to compare the results of CLIP-OGD to alternative experimental designs which are not made for Neyman allocation. Indeed, we find that the alternative designs incur a high variance, relative to CLIP-OGD and the Neyman variance.

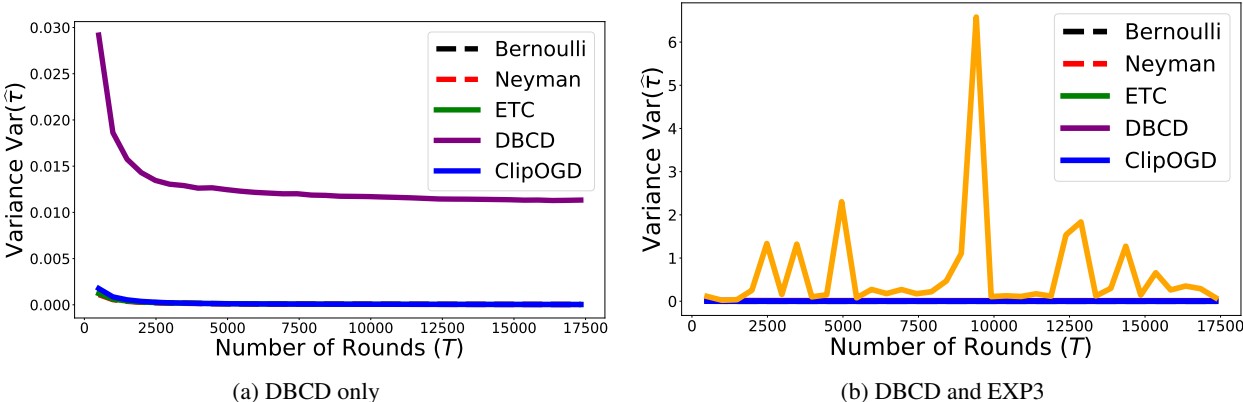

(a) DBCD only

(b) DBCD and EXP3

Figure 4: Comparison of (Unnormalized) Variances:

In Figure 4, we plot the variance of the adaptive designs on an unnormalized scale. We include "Doubly Biased Coin Design" proposed by Eisele [1994] as DBCD in Fig 4a and both DBCD and EXP3 in Fig 4b. We find that both DBCD and EXP3 suffer from higher variance. This is because they are not designed for Adaptive Neyman Allocation as defined in this paper: DBCD targets a different allocation rule and EXP3 minimizes outcome regret so that essentially only one arm is pulled. Both of these algorithms let the sampling probabilities $P_t$ get too close to the boundary of $[0, 1]$, resulting in excessively large variance.

# B  General Analysis of Adaptive Neyman Allocation

In this section, we provide general analysis relevant for the problem of Adaptive Neyman Allocation. In Section B.1, we analyze the adaptive Horvitz–Thompson estimator. In Section B.2, we derive the optimal non-adaptive Neyman design in terms of the potential outcomes. In Section B.3, we show the equivalence of Neyman ratio and expected Neyman regret. For completeness, all propositions re-appear in the appendix.

## B.1  Analysis of Adaptive Horvitz–Thompson Estimator (Propositions 2.1 and 2.2)

Throughout these proofs, we break up the adaptive Horvitz–Thompson estimator into the sum of individual estimators. For each $t \in [T]$, define

$$\hat{\tau}_t = Y_t\Big(\frac{\mathbf{1}[Z_t = 1]}{P_t} - \frac{\mathbf{1}[Z_t = 0]}{1 - P_t}\Big)$$

so that the sequential Horvitz–Thompson estimator is equal to $\hat{\tau} = (1/T)\sum_{t=1}^{T} \hat{\tau}_t$. This mirrors how the average treatment effect is the average of individual treatment effects, i.e. $\tau = (1/T)\sum_{t=1}^{T} \tau_t$.

We begin by proving Proposition 2.1, which establishes the unbiasedness of the adaptive Horvitz–Thompson estimator, subject to a positivity condition.

**Proposition 2.1.** *If* $\min\{P_t, 1 - P_t\} > 0$ *almost surely for all* $t \in [T]$ *then the adaptive Horvitz–Thompson estimator is unbiased:* $\mathbb{E}[\hat{\tau}] = \tau$.

*Proof.* Observe that by linearity of expectation, we can break the expectation of the adaptive Horvitz–Thompson estimator as

$$\mathbb{E}[\hat{\tau}] = \frac{1}{T}\sum_{t=1}^{T} \mathbb{E}[\hat{\tau}_t] \ .$$

Thus, it suffices to show that the individual effect estimators are unbiased: $\mathbb{E}[\hat{\tau}_t] = \tau_t$. Observe that if the positivity condition holds, then we have that the conditional expectation may be computed as

$$\begin{aligned}
\mathbb{E}[\hat{\tau}_t \mid \mathcal{F}_t] &= \mathbb{E}\Big[Y_t\Big(\frac{\mathbf{1}[Z_t = 1]}{P_t} - \frac{\mathbf{1}[Z_t = 0]}{1 - P_t}\Big) \mid \mathcal{F}_t\Big] \\
&= P_t \cdot \Big(\frac{y_t(1)}{P_t}\Big) + (1 - P_t) \cdot \Big(\frac{y_t(0)}{1 - P_t}\Big) \\
&= y_t(1) - y_t(0) \\
&= \tau_t \ .
\end{aligned}$$

The result follows by iterated expectation, $\mathbb{E}[\hat{\tau}_t] = \mathbb{E}[\mathbb{E}[\hat{\tau}_t \mid \mathcal{F}_t]] = \tau_t$. $\qquad\square$

Next, we prove Proposition 2.2, which derives the variance of the adaptive Horvitz–Thompson estimator.

**Proposition 2.2.** *The variance of the adaptive Horvitz–Thompson estimator is*

$$T \cdot \mathrm{Var}(\hat{\tau}) = \frac{1}{T}\sum_{t=1}^{T}\Big(y_t(1)^2 \,\mathbb{E}\Big[\frac{1}{P_t}\Big] + y_t(0)^2 \,\mathbb{E}\Big[\frac{1}{1 - P_t}\Big]\Big) - \frac{1}{T}\sum_{t=1}^{T} \tau_t^2 \ .$$

*Proof.* We begin by decomposing the variance of the adaptive Horvitz–Thompson estimator as

$$\mathrm{Var}(\hat{\tau}) = \mathrm{Var}\Big(\frac{1}{T}\sum_{t=1}^{T} \hat{\tau}_t\Big) = \frac{1}{T^2}\sum_{t=1}^{T}\sum_{s=1}^{T} \mathrm{Cov}(\hat{\tau}_t, \hat{\tau}_s) \ .$$

We now aim to compute each of these individual covariance terms. Before continuing, observe that, by construction, the individual effect estimators are conditionally unbiased $\mathbb{E}[\hat{\tau}_t \mid \mathcal{F}_t] = \tau_t$. It follows by iterated expectation that the individual effect estimators are unbiased (unconditionally), i.e. $\mathbb{E}[\hat{\tau}_t] = \tau_t$. Suppose that $s > t$. In this case, the covariance between the individual estimators is equal to zero as,

$$\begin{aligned}
\mathrm{Cov}(\hat{\tau}_t, \hat{\tau}_s) &= \mathbb{E}[\hat{\tau}_t \hat{\tau}_s] - \mathbb{E}[\hat{\tau}_t]\,\mathbb{E}[\hat{\tau}_s] \\
&= \mathbb{E}[\hat{\tau}_t\,\mathbb{E}[\hat{\tau}_s \mid \mathcal{F}_s]] - \tau_t \tau_s \\
&= \tau_s\,\mathbb{E}[\hat{\tau}_t] - \tau_t \tau_s
\end{aligned}$$

$$= \tau_s \tau_t - \tau_t \tau_s$$
$$= 0 \ .$$

Now let us compute the variance of an individual effect estimator. Observe that the variance may be decomposed as

$$\mathrm{Var}(\hat{\tau}_t) = \mathbb{E}[\hat{\tau}_t^2] - \mathbb{E}[\hat{\tau}_t]^2 \ .$$

Because the individual estimator is unbiased, we have that $\mathbb{E}[\hat{\tau}_t]^2 = \tau_t^2$. Let us now analyze the first term.

$$\mathbb{E}[\hat{\tau}_t^2] = \mathbb{E}\big[\mathbb{E}[\hat{\tau}_t^2 \mid \mathcal{F}_t]\big] \qquad\qquad\qquad \text{(iterated expectation)}$$
$$= \mathbb{E}\Big[\mathbb{E}\Big[Y_t^2\Big(\frac{\mathbf{1}[Z_t = 1]}{P_t^2} - \frac{\mathbf{1}[Z_t = 0]}{(1 - P_t)^2}\Big) \mid \mathcal{F}_t\Big]\Big]$$
$$= \mathbb{E}\Big[y_t(1)^2 \frac{1}{P_t} + y_t(0)^2 \frac{1}{1 - P_t}\Big]$$
$$= y_t(1)^2 \cdot \mathbb{E}\Big[\frac{1}{P_t}\Big] + y_t(0)^2 \cdot \mathbb{E}\Big[\frac{1}{1 - P_t}\Big] \ .$$

Thus, this establishes that the variance of an individual estimator is equal to

$$\mathrm{Var}(\hat{\tau}_t) = y_t(1)^2 \cdot \mathbb{E}\Big[\frac{1}{P_t}\Big] + y_t(0)^2 \cdot \mathbb{E}\Big[\frac{1}{1 - P_t}\Big] - \tau_t^2 \ .$$

Combining terms, we have that the variance of the adaptive Horvitz–Thompson estimator is

$$T \cdot \mathrm{Var}(\hat{\tau}) = \frac{1}{T}\sum_{t=1}^{T}\sum_{s=1}^{T}\mathrm{Cov}(\hat{\tau}_t, \hat{\tau}_s)$$
$$= \frac{1}{T}\sum_{t=1}^{T}\mathrm{Var}(\hat{\tau}_t)$$
$$= \frac{1}{T}\sum_{t=1}^{T}\Big(y_t(1)^2 \cdot \mathbb{E}\Big[\frac{1}{P_t}\Big] + y_t(0)^2 \cdot \mathbb{E}\Big[\frac{1}{1 - P_t}\Big]\Big) - \frac{1}{T}\sum_{t=1}^{T}\tau_t^2 \qquad \square$$

## B.2   Derivation of the Neyman Design (Proposition 3.1)

In this section, we prove Proposition 3.1 which derives the (infeasible) non-adaptive Neyman design in terms of the Neyman probability $p^*$ and corresponding Neyman variance $V_N$. We also show that, under Assumption 1, the Neyman variance achieves the parametric rate.

**Proposition 3.1.** *The Neyman variance is $T \cdot V_N = 2(1 + \rho)S(1)S(0)$, which is achieved by the Neyman probability $p^* = (1 + S(0)/S(1))^{-1}$.*

*Proof.* Using Proposition 2.2, we have that the variance of the (non-adaptive) Bernoulli design with probability $p \in (0, 1)$ is equal to

$$T \cdot V_p = S(1)^2\Big(\frac{1}{p} - 1\Big) + S(0)^2\Big(\frac{1}{1 - p} - 1\Big) + 2\rho S(1)S(0) \ .$$

Thus, the optimal Neyman design is obtained by the $p^*$ which minimizes the above. The first order condition stipulates that

$$\frac{\partial}{\partial p}\big[T \cdot V_p\big]\Big|_{p=p*} = 0 \Leftrightarrow -S(1)^2\Big(\frac{1}{p^*}\Big)^2 + S(0)^2\Big(\frac{1}{1 - p^*}\Big)^2 = 0 \ ,$$

which is solved by $p^* = \big(1 + S(0)/S(1)\big)^{-1}$. Substituting this $p^*$ back into the variance yields the Neyman variance:

$$T \cdot V_{p*} = S(1)^2\Big(\frac{1}{p^*} - 1\Big) + S(0)^2\Big(\frac{1}{1 - p^*} - 1\Big) + 2\rho S(1)S(0)$$
$$= S(1)^2 \cdot \frac{S(0)}{S(1)} + S(0)^2 \cdot \frac{S(1)}{S(0)} + 2\rho S(1)S(0)$$
$$= 2(1 + \rho)S(1)S(0) \ . \qquad \square$$

Next, we show that under Assumption 1, the Neyman variance achieves the parametric rate.

**Proposition B.1.** *Under Assumption 1, the Neyman variance achieves the parametric rate: $V_N = \Theta(1/T)$.*

*Proof.* Proposition 3.1, derives the Neyman variance: $T \cdot V_N = 2(1 + \rho)S(1)S(0)$.

We begin by showing that the Neyman variance is asymptotically bounded from below. Moreover, Assumption 1 stipulates that there exists a constant $c > 0$ which lower bounds the second moments as $S(1) \geqslant c$ and $S(0) \geqslant c$ and the correlation as $(1 + \rho) \geqslant c$. Thus, the normalized Neyman variance is bounded below as $T \cdot V_N \geqslant 2c^3$.

Next, we show that the Neyman variance is asymptotically bounded from above at the same rate. Assumption 1 stipulates that there exists a constant $C > 0$ which upper bounds the second moments as $S(1) \leqslant C$ and $S(0) \leqslant C$. The correlation is bounded between $\rho \in [-1, 1]$ so that $(1 + \rho) \leqslant 2$. These bounds together yield that the normalized Neyman variance is bounded above as $T \cdot V_N \leqslant 4C^2$.

Together, these bounds establish that, under Assumption 1, we have that $V_N = \Theta(1/T)$. $\square$

## B.3 Equivalence of Neyman Ratio and Neyman Regret (Theorem 4.1)

In this section, we prove Theorem 4.1, which demonstrates the equivalence between the Neyman Ratio and the expected Neyman regret.

**Theorem 4.1.** *Under Assumption 1, the Neyman ratio is within a constant factor of the $1/T$-scaled expected Neyman regret: $\kappa_T = \Theta(\frac{1}{T} \mathbb{E}[\mathcal{R}_T])$.*

*Proof.* Recall that the Neyman ratio is defined as

$$\kappa_T = \frac{V - V_N}{V_N} = \frac{T \cdot V - T \cdot V_N}{T \cdot V_N} \ ,$$

where the second equality follows by multiplying the numerator and the denominator by $T$. Observe that by Proposition 2.2, the numerator is given by

$$T \cdot V - T \cdot V_N = \frac{1}{T} \sum_{t=1}^{T} \left( y_t(1)^2 \cdot \mathbb{E}\left[\frac{1}{P_t}\right] + y_t(0)^2 \cdot \mathbb{E}\left[\frac{1}{1 - P_t}\right] \right)$$

$$- \min_{p^* \in [0,1]} \frac{1}{T} \sum_{t=1}^{T} \left( y_t(1)^2 \cdot \frac{1}{p^*} + y_t(0)^2 \cdot \frac{1}{1 - p^*} \right)$$

$$= \mathbb{E}\left[\frac{1}{T} \sum_{t=1}^{T} f_t(P_t)\right] - \min_{p^* \in [0,1]} \frac{1}{T} \sum_{t=1}^{T} f_t(p^*)$$

$$= \frac{1}{T} \mathbb{E}\left[\sum_{t=1}^{T} f_t(P_t) - \min_{p^* \in [0,1]} \sum_{t=1}^{T} f_t(p^*)\right]$$

$$= \frac{1}{T} \mathbb{E}[\mathcal{R}_T] \ .$$

Proposition B.1 shows that under Assumption 1, $T \cdot V_N = \Theta(1)$ so that the denominator is asymptotically constant. Thus, we have that $\kappa_T = \Theta(\frac{1}{T} \mathbb{E}[\mathcal{R}_T])$. $\square$

## C Analysis of Neyman Regret for CLIP-OGD

In this section, we will prove Theorem 4.2, which establish that CLIP-OGD achieves $\mathcal{O}(\sqrt{T \log T})$ expected Neyman regret under our assumptions on the potential outcomes. While the main paper used capital letters $P_t$ and $G_t$ to signify that the treatment probability and gradient estimator were random variables, we use lower case letters $p_t$ and $g_t$ in the appendix for the purposes of more aesthetically appealing proofs. Throughout the analysis, we define $\Delta_t = [\delta_t, 1 - \delta_t]$ and $a = 1 + C/c$ for notational convenience.

### C.1 Proof of Theorem 4.2

The first lemma is a bound on the distance of treatment probability $p_{t+1}$ to the optimal $p^*$ in terms of the previous treatment probability, gradient estimate, and whether the projection interval contains the optimal $p^*$.

**Lemma C.1.** *For each iteration $t \in [T]$,*

$$|p_{t+1} - p^*| \leqslant |(p_t - \eta g_t) - p^*| + \delta_t \mathbf{1}[p^* \notin \Delta_t] \ .$$

*Proof.* If $p^\star \in \Delta_t$, then the statement holds by Pythagorean theorem. Otherwise, note that the most that the projection operation onto $\Delta_t$ can move a point is exactly $\delta_t$. Therefore, the distance between $\mathcal{P}_{\delta_t}(p_t - \eta g_t)$ and $p^\star$ is at most $\delta_t$ larger than that between $p_t - \eta g_t$ and $p^\star$. $\qquad\square$

Next, we show that Assumption 1 implies that the optimal Neyman probability lies within an interval bounded away from zero.

**Lemma C.2.** *Under Assumption 1, $p^* \in [\frac{1}{a}, 1 - \frac{1}{a}]$, where $a = 1 + C/c \geqslant 2$.*

*Proof.* As shown previously, the optimal Neyman probability is equal to

$$p^* = \left(1 + \sqrt{\frac{\sum_{t=1}^{T} y_t(0)^2}{\sum_{t=1}^{T} y_t(1)^2}}\right)^{-1}$$

Recall that Assumption 1 places the following moment conditions on the potential outcomes:

$$c \leqslant \left(\frac{1}{T} \sum_{t=1}^{T} y_t(1)^2\right)^{1/2} \leqslant C \quad \text{and} \quad c \leqslant \left(\frac{1}{T} \sum_{t=1}^{T} y_t(1)^2\right)^{1/2} \leqslant C \ .$$

This bounds $p^*$ by

$$\left(1 + \frac{C}{c}\right)^{-1} \leqslant p^* \leqslant \left(1 + \frac{c}{C}\right)^{-1} \ .$$

The result follows by using the definition of $a = 1 + C/c$ to deduce that $(1/a) = (1 + C/c)^{-1}$ and $(1 - 1/a) = (1 + c/C)^{-1}$. $\qquad\square$

The next lemma guarantees that after a fixed number of iterations, the projection interval will contain the Neyman optimal $p^*$.

**Lemma C.3.** *We have that $p^* \in \Delta_t$ for all $t \geqslant (a/2)^\alpha$.*

*Proof.* Lemma C.2 guarantees that $p^* \in [1/a, 1 - 1/a]$, where $a = 1 + C/c$. Thus, $p^* \in \Delta_t$ if $\delta_t \leqslant 1/a$. Using the definition of $\delta_t$ and rearranging terms, we have that

$$\delta_t \leqslant 1/a \Leftrightarrow (1/2)t^{-1/\alpha} \leqslant 1/a \Leftrightarrow t \geqslant (a/2)^\alpha \ . \qquad\square$$

The next lemma bounds the expected difference between the cost objective $f_t$ evaluated at $p_t$ and the cost objective $f_t$ evaluated at the Neyman optimal probability $p^*$.

**Lemma C.4.** *For each iteration $t \in [T]$, we have the bound*

$$2\,\mathbb{E}\Big[f_t(p_t) - f_t(p^*)\Big] \leqslant \frac{1}{\eta}\Big[\mathbb{E}\big[(p_t - p^*)^2\big] - \mathbb{E}\big[(p_{t+1} - p^*)^2\big]\Big] + \eta\,\mathbb{E}[g_t^2] + 4\mathbf{1}\Big[t < \Big(\frac{a}{2}\Big)^\alpha\Big]\Big(\frac{\delta_t}{\eta} + \frac{\delta_t}{2}\,\mathbb{E}\big[|g_t|\big]\Big) \ .$$

*Proof.* Fix an iteration $t \in [T]$. By Lemma C.1, and using the triangle inequality, we have that

$$
\begin{aligned}
(p_{t+1} - p^*)^2 &\leqslant \left(|(p_t - \eta g_t) - p^*| + \delta_t \mathbf{1}[p^* \notin \Delta_t]\right)^2 \\
&= \left((p_t - \eta g_t) - p^*\right)^2 + \delta_t^2 \mathbf{1}[p^* \notin \Delta_t]^2 + 2|(p_t - \eta g_t) - p^*|\delta_t \mathbf{1}[p^* \notin \Delta_t] \\
&= \left((p_t - p^*) - \eta g_t\right)^2 + \mathbf{1}[p^* \notin \Delta_t]\left(\delta_t^2 + 2\delta_t \cdot |(p_t - p^*) - \eta g_t|\right) \\
&\leqslant \left((p_t - p^*) - \eta g_t\right)^2 + 2\delta_t \mathbf{1}[p^* \notin \Delta_t]\left(\delta_t + |p_t - p^*| + \eta|g_t|\right)
\end{aligned}
$$

Because $p_t \in [\delta_t, 1 - \delta_t]$, we have that $|p_t - p^*| \leqslant 1 - \delta_t$ so that

$$\leqslant \left((p_t - p^*) - \eta g_t\right)^2 + 2\delta_t \mathbf{1}[p^* \notin \Delta_t]\left(1 + \eta|g_t|\right)$$

$$=\leqslant (p_t - p^*)^2 + \eta^2 g_t^2 - 2\eta g_t (p_t - p^*) + 4\eta \cdot \mathbf{1}[p^* \notin \Delta_t]\Big(\frac{\delta_t}{\eta} + \frac{\delta_t}{2}|g_t|\Big)$$

.

Rearranging terms yields that

$$2\eta g_t (p_t - p^*) \leqslant \Big[(p_t - p^*)^2 - (p_{t+1} - p^*)^2\Big] + \eta^2 g_t^2 + 4\eta \cdot \mathbf{1}[p^* \notin \Delta_t]\Big(\frac{\delta_t}{\eta} + \frac{\delta_t}{2}|g_t|\Big) \ .$$

Dividing both sides by the step size $2\eta$ yields

$$g_t (p_t - p^*) \leqslant \frac{1}{2\eta}\Big[(p_t - p^*)^2 - (p_{t+1} - p^*)^2\Big] + \frac{\eta}{2}g_t^2 + 2\mathbf{1}[p^* \notin \Delta_t]\Big(\frac{\delta_t}{\eta} + \frac{\delta_t}{2}|g_t|\Big) \ .$$

Using convexity of $f_t$, adding and subtracting terms, and using the above, we have that

$$
\begin{aligned}
f_t(p_t) - f_t(p^*) &\leqslant \langle \nabla f_t(p_t), p_t - p^* \rangle & \text{(convexity)}\\
&= \langle g_t, p_t - p^* \rangle + \langle \nabla f_t(p_t) - g_t, p_t - p^* \rangle & \text{(adding, subtracting)}\\
&\leqslant \frac{1}{2\eta}\Big[(p_t - p^*)^2 - (p_{t+1} - p^*)^2\Big] + \frac{\eta}{2}g_t^2 + 2\mathbf{1}[p^* \notin \Delta_t]\Big(\frac{\delta_t}{\eta} + \frac{\delta_t}{2}|g_t|\Big) & \text{(above)}\\
&\quad + \langle \nabla f_t(p_t) - g_t, p_t - p^* \rangle
\end{aligned}
$$

By construction, we have that the gradient estimator is unbiased conditioned on $D_t$, i.e. $\mathbb{E}[g_t \mid D_t] = \nabla f_t(p_t)$. Thus, by iterated expectation we have that the gradient estimator is unbiased, i.e.

$$\mathbb{E}[\nabla f_t(p_t) - g_t] = \mathbb{E}[\mathbb{E}[\nabla f_t(p_t) - g_t \mid D_t]] = 0 \ .$$

Thus, taking expectations of both sides and applying Lemma C.3 yields

$$
\begin{aligned}
2\,\mathbb{E}\Big[f_t(p_t) - f_t(p^*)\Big] & \\
&\leqslant \frac{1}{\eta}\Big[\mathbb{E}\big[(p_t - p^*)^2\big] - \mathbb{E}\big[(p_{t+1} - p^*)^2\big]\Big] + \eta\,\mathbb{E}\big[g_t^2\big] + 4 \cdot \mathbf{1}[p^* \notin \Delta_t]\Big(\frac{\delta_t}{\eta} + \frac{\delta_t}{2}\,\mathbb{E}\big[|g_t|\big]\Big)\\
&\leqslant \frac{1}{\eta}\Big[\mathbb{E}\big[(p_t - p^*)^2\big] - \mathbb{E}\big[(p_{t+1} - p^*)^2\big]\Big] + \eta\,\mathbb{E}\big[g_t^2\big] + 4 \cdot \mathbf{1}\Big[t < \Big(\frac{a}{2}\Big)^\alpha\Big]\Big(\frac{\delta_t}{\eta} + \frac{\delta_t}{2}\,\mathbb{E}\big[|g_t|\big]\Big) \qquad \square
\end{aligned}
$$

The following lemma derives bounds on the first and second (raw) moments of the gradient estimator at each iteration.

**Lemma C.5.** *For each $t \in [T]$, the gradient estimates have bounded first and second moments:*

$$\mathbb{E}\Big[g_t^2\Big] \leqslant 2^5 t^{5/\alpha} \cdot \big(y_t(1)^4 + y_t(0)^4\big)$$

$$\mathbb{E}\Big[|g_t|\Big] \leqslant 2^2 t^{2/\alpha} \cdot \big(y_t(1)^2 + y_t(0)^2\big)$$

*Proof.* We begin by handling the $\mathbb{E}\Big[g_t^2\Big]$ term. By definition of the gradient estimator, we have that the conditional expectation is at most

$$
\begin{aligned}
\mathbb{E}[g_t^2 \mid D_t] &= p_t \cdot \Big(\frac{y_t(1)^2}{p_t^3}\Big)^2 + (1 - p_t) \cdot \Big(\frac{y_t(0)^2}{(1 - p_t)^3}\Big)^2\\
&= \frac{y_t(1)^4}{p_t^5} + \frac{y_t(0)^4}{(1 - p_t)^5}
\end{aligned}
$$

By definition of the algorithm, we have that $p_t \in [\delta_t, 1 - \delta_t]$ at iteration $t$. Thus, we may invoke the bound:

$$
\begin{aligned}
&\leqslant \delta_t^{-5}\Big(y_t(1)^4 + y_t(1)^4\Big)\\
&= [(1/2)t^{-1/\alpha}]^{-5} \cdot \Big(y_t(1)^4 + y_t(1)^4\Big)\\
&= 2^5 t^{5/\alpha} \cdot \Big(y_t(1)^4 + y_t(1)^4\Big) \ ,
\end{aligned}
$$

and the desired bound on $\mathbb{E}[g_t^2]$ follows from applying the law of iterated expectation.

The bound on the $\mathbb{E}\Big[|g_t|\Big]$ term follows in a similar way. By definition of the gradient estimator, we have that the conditional expectation is at most

$$
\begin{aligned}
\mathbb{E}[|g_t| \mid D_t] &= p_t \cdot \left|\frac{y_t(1)^2}{p_t^3}\right| + (1 - p_t) \cdot \left|\frac{y_t(0)^2}{(1 - p_t)^3}\right| \\
&= \frac{y_t(1)^2}{p_t^2} + \frac{y_t(0)^2}{(1 - p_t)^2} \\
&\leqslant \delta_t^{-2}\Big(y_t(1)^2 + y_t(1)^2\Big) \\
&= [(1/2)t^{-1/\alpha}]^{-2} \cdot \Big(y_t(1)^2 + y_t(1)^2\Big) \\
&= 2^2 t^{2/\alpha} \cdot \Big(y_t(1)^2 + y_t(1)^2\Big) \ ,
\end{aligned}
$$

and the desired bound on $\mathbb{E}[|g_t|]$ follows from applying the law of iterated expectation. $\qquad \square$

The following proposition bounds the expected Neyman regret for general settings of the projection parameter $\alpha$.

**Proposition C.1.** *Suppose Assumption 1 holds. Then, for any choice of projection parameter $\alpha \geqslant 2$ (possibly depending on $T$) and for the step size $\eta = \sqrt{\frac{e^\alpha}{T^{1+5/\alpha}}}$, a finite-sample bound on the expected Neyman regret incurred by* CLIP-OGD *is*

$$
\mathbb{E}\big[\mathcal{R}_T\big] \leqslant (2^2 e^{a/2} + 2^5 C^4)\sqrt{e^\alpha T^{1+5/\alpha}} + 2^2 C^2 e^{2+a/2}\sqrt{e^\alpha T} \ .
$$

*Proof.* By Lemma C.4, we have that the regret is at most

$$
\begin{aligned}
2\,\mathbb{E}\big[\mathcal{R}_T\big] &= \sum_{t=1}^T 2\,\mathbb{E}\Big[f_t(p_t) - f_t(p^*)\Big] \\
&\leqslant \frac{1}{\eta}\sum_{t=1}^T \Big[\mathbb{E}\Big[(p_t - p^*)^2\Big] - \mathbb{E}\Big[(p_{t+1} - p^*)^2\Big]\Big] + \eta \sum_{t=1}^T \mathbb{E}[g_t^2] \\
&\quad + 4\sum_{t=1}^T \mathbf{1}\Big[t < \Big(\frac{a}{2}\Big)^\alpha\Big]\Big(\frac{\delta_t}{\eta} + \frac{\delta_t}{2}\,\mathbb{E}\Big[|g_t|\Big]\Big)
\end{aligned}
$$

Using a telescoping argument, we have that the first term is bounded by

$$
\frac{1}{\eta}\sum_{t=1}^T \Big[\mathbb{E}\Big[(p_t - p^*)^2\Big] - \mathbb{E}\Big[(p_{t+1} - p^*)^2\Big]\Big] \leqslant \frac{1}{\eta}\,\mathbb{E}\Big[(p_1 - p^*)^2\Big] \leqslant \frac{1}{\eta} \ .
$$

Using Lemma C.5 and Assumption 1, the sum in the second term may be bounded as

$$
\begin{aligned}
\sum_{t=1}^T \mathbb{E}[g_t^2] &\leqslant \sum_{t=1}^T 2^5 t^{5/\alpha}(y_t(1)^4 + y_t(0)^4) && \text{(Lemma C.5)} \\
&\leqslant 2^5 T^{5/\alpha}\Big(\sum_{t=1}^T y_t(1)^4 + \sum_{t=1}^T y_t(0)^4\Big) \\
&\leqslant 2^5 T^{5/\alpha}\Big(2C^4 T\Big) && \text{(Assumption 1)} \\
&= 2^6 C^4 T^{1+5/\alpha} \ .
\end{aligned}
$$

Next, we deal with the third term by breaking it up into two more terms. Define $t^* = \lceil (a/2)^\alpha \rceil$. The third term can be broken into two terms:

$$
4\sum_{t=1}^T \mathbf{1}\Big[t < \Big(\frac{a}{2}\Big)^\alpha\Big]\Big(\frac{\delta_t}{\eta} + \frac{\delta_t}{2}\,\mathbb{E}\Big[|g_t|\Big]\Big) = \frac{4}{\eta}\sum_{t=1}^{t^*-1} \delta_t + 2\sum_{t=1}^{t^*-1} \delta_t\,\mathbb{E}[|g_t|] \ .
$$

The first of these two terms can be bounded in the following way. Using that $\alpha \geqslant 2$ we have that

$$
\sum_{t=1}^{t^*-1} \delta_t = \sum_{t=1}^{t^*-1} (1/2)t^{-1/\alpha}
$$

$$= (1/2)\Big[1 + \sum_{t=2}^{t^*-1} t^{-1/\alpha}\Big]$$

$$\leqslant (1/2)\Big[1 + \int_{x=1}^{t^*-1} x^{-1/\alpha}dx\Big]$$

$$= (1/2)\Big[1 + \frac{\alpha}{\alpha-1}\cdot((t^*-1)^{(1-1/\alpha)}-1)\Big]$$

$$\leqslant (t^*-1)^{(1-1/\alpha)}$$

$$\leqslant (a/2)^{\alpha(1-1/\alpha)}$$

$$= (a/2)^{\alpha-1} \ .$$

The second term can be bounded using Lemma C.5 as follows:

$$\sum_{t=1}^{t^*-1} \delta_t \, \mathbb{E}[|g_t|] \leqslant \sum_{t=1}^{t^*-1} (1/2)t^{-1/\alpha}2^2 t^{2/\alpha}\big(y_t(1)^2 + y_t(0)^2\big) \qquad \text{(Lemma C.5)}$$

$$= 2\sum_{t=1}^{t^*-1} t^{1/\alpha}\big(y_t(1)^2 + y_t(0)^2\big)$$

$$\leqslant 2\Big(\sum_{t=1}^{t^*-1} t^{2/\alpha}\Big)^{1/2}\Big[\Big(\sum_{t=1}^{t^*-1} y_t(1)^4\Big)^{1/2} + \Big(\sum_{t=1}^{t^*-1} y_t(0)^4\Big)^{1/2}\Big] \ , \qquad \text{(Cauchy-Schwarz)}$$

where the last inequality follows from Cauchy–Schwarz. By extending the sum involving the outcomes to all units, we obtain an upper bound on which we can apply the bounded moment assumption:

$$\leqslant 2\Big(\sum_{t=1}^{t^*-1} t^{2/\alpha}\Big)^{1/2}\Big[\Big(\sum_{t=1}^{T} y_t(1)^4\Big)^{1/2} + \Big(\sum_{t=1}^{T} y_t(0)^4\Big)^{1/2}\Big]$$

$$\leqslant 2\Big(\sum_{t=1}^{t^*-1} t^{2/\alpha}\Big)^{1/2}\Big[2(C^4 T)^{1/2}\Big] \qquad \text{(Assumption 1)}$$

What remains in this step is to bound the first term above. By replacing each of the $t$ in the sum with $t^*$, we obtain the following upper bound:

$$\leqslant 2(t^*-1)^{(1/2+1/\alpha)}\Big[2(C^4 T)^{1/2}\Big]$$

$$\leqslant 2((a/2)^\alpha)^{(1/2+1/\alpha)}\Big[2(C^4 T)^{1/2}\Big]$$

$$= 2^2 C^2 \Big(\frac{a}{2}\Big)^{(1+\alpha/2)} T^{1/2} \ .$$

Using the above work, we have that the overall regret is bounded as

$$2\,\mathbb{E}\Big[\mathcal{R}_T\Big] \leqslant \frac{1}{\eta} + \eta 2^6 C^4 T^{1+5/\alpha} + \frac{4}{\eta}\Big(\frac{a}{2}\Big)^{(\alpha-1)} + 2^3 C^2 \Big(\frac{a}{2}\Big)^{(1+\alpha/2)} T^{1/2}$$

$$= \frac{1}{\eta}\Big(1 + 4\Big(\frac{a}{2}\Big)^{(\alpha-1)}\Big) + \eta 2^6 C^4 T^{1+5/\alpha} + 2^3 C^2 \Big(\frac{a}{2}\Big)^{(1+\alpha/2)} T^{1/2} \ .$$

Next, we separate the constant $a$ from the projection parameter $\alpha$. To this end, we use the inequality $y^r \leqslant e^{1+y}\cdot e^r$ for all $y \in \mathbb{R}$ and $r \geqslant 0$, to obtain

$$\Big(\frac{a}{2}\Big)^{(\alpha-1)} \leqslant e^{1+a/2}e^{\alpha-1} = e^{a/2}e^\alpha \quad \text{and} \quad \Big(\frac{a}{2}\Big)^{(1+\alpha/2)} \leqslant e^{1+a/2}e^{1+\alpha/2} = e^{2+a/2}e^{\alpha/2} \ ,$$

where we have used that $a \geqslant 2$. Substituting these back into the above, we have that the regret bound is Substituting this quantities into the above, we have that the regret bound is

$$2\,\mathbb{E}[\mathcal{R}_T] \leqslant \frac{1}{\eta}\Big(1 + 4e^{a/2}e^\alpha\Big) + \eta 2^6 C^4 T^{1+5/\alpha} + 2^3 C^2 e^{2+a/2}e^{\alpha/2} T^{1/2}$$

$$\leqslant \frac{1}{\eta} 2^3 e^{a/2} e^{\alpha} + \eta 2^6 C^4 T^{1+5/\alpha} + 2^3 C^2 e^{2+a/2} e^{\alpha/2} T^{1/2}$$

By setting $\eta = \sqrt{\frac{e^{\alpha}}{T^{1+5/\alpha}}}$ to minimize the bound above, we have that the expected Neyman regret is bounded as

$$= (2^3 e^{a/2} + 2^6 C^4)\sqrt{e^{\alpha} T^{1+5/\alpha}} + 2^3 C^2 e^{2+a/2}\sqrt{e^{\alpha} T} .$$

where we have used that and the result follows by dividing both sides by 2. $\qquad\square$

Proposition C.1 demonstrates that many different values of $\alpha$ will guarantee sublinear expected Neyman regret. For example, setting $\alpha$ to be a constant satisfying $\alpha > 5$ will ensure sublinear expected Neyman regret. However, by tuning $\alpha$ according to the analysis above, we can achieve $\mathcal{O}(\sqrt{T \log(T)})$ expected Neyman regret, as demonstrated by Theorem 4.2.

**Theorem 4.2\*.** *Under Assumption 1 the parameter values $\eta = \sqrt{1/T}$ and $\alpha = \sqrt{5 \log(T)}$ ensure the expected Neyman regret of* CLIP-OGD *is bounded as*

$$\mathbb{E}\big[\mathcal{R}_T\big] \leqslant \left(2^2 e^{a/2} + 2^5 C^4 + 2^2 C^2 e^{2+a/2}\right) \cdot \sqrt{T} \cdot \exp(\sqrt{5 \log(T)}) ,$$

*which implies that $\mathbb{E}\big[\mathcal{R}_T\big] \leqslant \tilde{\mathcal{O}}(\sqrt{T})$.*

*Proof.* Observe that for $\alpha = \sqrt{5 \log(T)}$, we have that the step size posited in Proposition C.1 (i.e. $\eta = \sqrt{\frac{e^{\alpha}}{T^{1+5/\alpha}}}$) is equal to $\eta = \sqrt{1/T}$. Thus, by rearranging terms and using the result of Proposition C.1, we have that expected Neyman regret is bounded as

$$\mathbb{E}\big[\mathcal{R}_T\big] \leqslant (2^2 e^{a/2} + 2^5 C^4)\sqrt{e^{\alpha} T^{1+5/\alpha}} + 2^2 C^2 e^{2+a/2}\sqrt{e^{\alpha} T}$$
$$= (2^2 e^{a/2} + 2^5 C^4)\sqrt{T} \cdot \sqrt{e^{\alpha} T^{5/\alpha}} + 2^2 C^2 e^{2+a/2}\sqrt{T} \cdot \sqrt{e^{\alpha}} .$$

The difficulty is now to find which setting of $\alpha$ will make this bound smallest. Observe that the real tension is in the first term and we can re-write the relevant part of this term as

$$\sqrt{e^{\alpha} T^{5/\alpha}} = \left[\exp(\alpha + \log(T^{5/\alpha}))\right]^{1/2} = \left[\exp(\alpha + (5/\alpha)\log(T))\right]^{1/2} .$$

To minimize this term, we select $\alpha = \sqrt{5 \log(T)}$ which results in

$$\sqrt{e^{\alpha} T^{5/\alpha}} = \exp(\sqrt{5 \log(T)}) .$$

Likewise, this choice of $\alpha$ results in $\sqrt{e^{\alpha}} \leqslant e^{\alpha} = \exp(\sqrt{5 \log(T)})$. Putting this together yields the desired finite sample regret bound:

$$\mathbb{E}\big[\mathcal{R}_T\big] \leqslant \left(2^2 e^{a/2} + 2^5 C^4 + 2^2 C^2 e^{2+a/2}\right) \cdot \sqrt{T} \cdot \exp(\sqrt{5 \log(T)}) .$$

The result follows by observing that the terms inside the parenthesis are constant by Assumption 1 and the function $\exp(\sqrt{5 \log(T)})$ is subpolynomial. $\qquad\square$

## C.2 Selecting Parameters When Moment Bounds are Known

We briefly remark on how to select the step size parameter $\eta$ when the experimenter can correctly specify the constants $C \geqslant c$ used in the Assumption 1. The proof of Proposition C.1 shows that for general parameters, the Neyman regret may be bounded as

$$\mathbb{E}[\mathcal{R}_T] \leqslant \frac{1}{\eta} 2^2 e^{a/2} e^{\alpha} + \eta 2^5 C^4 T^{1+5/\alpha} + 2^2 C^2 e^{2+a/2} e^{\alpha/2} T^{1/2} .$$

To optimize the step size with respect to these constants, one would choose $\alpha = \sqrt{5 \log(T)}$ and

$$\eta = \frac{e^{\frac{1}{4} \cdot (1 + C/c)}}{2\sqrt{2} C^2} \cdot \frac{1}{\sqrt{T}} ,$$

where we have used that $a = 1 + C/c$. When these moment bounds are correctly specified, this choice of step size will likely yield improved convergence rates, as our bound on the Neyman regret will have a factor of $C^2$ rather than $C^4$.

# D Analysis for Inference in Large Samples

In this section, we provide the necessary statistical tools for constructing asymptotically valid confidence intervals. This can be done in two main steps. First, in Section D.1 we construct a conservative variance estimator which we show is consistent in probability. Then, in Section D.2, we show that the resulting Chebyshev-type intervals are asymptotically valid.

While the main paper used capital letters $P_t$ and $G_t$ to signify that the treatment probability and gradient estimator were random variables, we use lower case letters $p_t$ and $g_t$ in the appendix for the purposes of more aesthetically appealing proofs. Throughout the analysis, we use the parameter settings $\eta = \sqrt{1/T}$ and $\alpha = \sqrt{5\log(T)}$, which are recommended in the main paper. However, we suspect that many of our results will go through for the class of parameters $\eta = \sqrt{\frac{e^\alpha}{T^{1+5/\alpha}}}$ and $\alpha > 5$ which appear in Proposition C.1.

The following lemma shows that under Assumptions 1 and 2, the variance of the adaptive Horvitz–Thompson estimator under CLIP-OGD achieves the parametric rate.

**Lemma D.1.** *Assumptions 1 and 2, the variance of the adaptive Horvitz–Thompson estimator under* CLIP-OGD *achieves the parametric rate:* $\mathrm{Var}(\hat\tau) = \Theta(1/T)$.

*Proof.* Theorem 4.1 shows that under Assumption 1, they Neyman ratio is order equivalent to the $1/T$-scaled expected Neyman regret, i.e. $\kappa_T = \Theta((1/T)\,\mathbb{E}[\mathcal{R}_T])$. Theorem 4.2 shows that under these assumptions, CLIP-OGD achieves sublinear expected Neyman regret $\mathbb{E}[\mathcal{R}_T] = o(T)$ which implies that $\limsup \kappa_T \leqslant 0$. Likewise, Assumption 2 states that the negative expected Neyman regret is sublinear, $-\mathbb{E}[\mathcal{R}_T] = o(T)$, which implies that $\liminf \kappa_T \geqslant 0$. Thus, we have that the Neyman ratio converges to zero, e.g. $\lim \kappa_T = 0$.

By recalling the definition of the Neyman ratio, we have that

$$0 = \lim_{T\to\infty} \kappa_T = \lim_{T\to\infty} \frac{V - V_{\mathrm{N}}}{V_{\mathrm{N}}} = \lim_{T\to\infty} \frac{T \cdot V - T \cdot V_{\mathrm{N}}}{T \cdot V_{\mathrm{N}}} \ .$$

Proposition B.1 demonstrates that $T \cdot V_{\mathrm{N}} = \Theta(1)$. Together with the above, this implies that $T \cdot V = \Theta(1)$. $\qquad\square$

The following lemma shows that under the recommended parameter settings, the (random) treatment probabilities are bounded away from zero and one.

**Lemma D.2.** *When* $\alpha = \sqrt{5\log(T)}$, *we have that for all iterations* $t \in [T]$, *the inverse of projection parameter is bounded:*

$$\frac{1}{\delta_t} \leqslant 2\exp(\sqrt{\log(T^{1/5})}) = \tilde{\mathcal{O}}(1) \ .$$

*Proof.* A uniform upper bound on the inverse of the projection parameters is

$$\frac{1}{\delta_t} \leqslant \frac{1}{\delta_T} = \frac{1}{(1/2)T^{-1/\alpha}} = 2T^{1/\alpha} = 2\exp(\tfrac{1}{\alpha}\log(T)) = 2\exp(\sqrt{\log(T^{1/5})}) \ .$$

To complete the proof, observe that the function $h(T) = \exp(\sqrt{1/5} \cdot \sqrt{\log(T)})$ is subpolynomial, so that we can write it as $\tilde{\mathcal{O}}(1)$. $\qquad\square$

## D.1 Conservative Variance Estimator (Theorem 5.1)

In this section, we prove Theorem 5.1 which establishes that the normalized variance estimator is converges in probability to the normalized variance upper bound at a $\tilde{\mathcal{O}}_p(T^{-1/2})$ rate. Before continuing, let us review the relevant quantities. Recall that the Neyman variance and the corresponding upper bound are given by

$$T \cdot V_{\mathrm{N}} = 2(1+\rho)S(1)S(0) \quad \text{and} \quad T \cdot \mathrm{VB} = 4S(1)S(0) \ ,$$

where the second moments $S(1)$ and $S(0)$ are defined as

$$S(1)^2 = \frac{1}{T}\sum_{t=1}^{T} y_t(1)^2 \quad \text{and} \quad S(0)^2 = \frac{1}{T}\sum_{t=1}^{T} y_t(0)^2 \ .$$

Our variance estimator is defined as

$$T \cdot \widehat{\mathrm{VB}} = \sqrt{\widehat{A(1)} \cdot \widehat{A(0)}} \quad \text{where} \quad \widehat{A(1)} = \frac{1}{T} \sum_{t=1}^{T} y_t(1)^2 \frac{\mathbf{1}[Z_t = 1]}{p_t} \quad \text{and} \quad \widehat{A(0)} = \frac{1}{T} \sum_{t=1}^{T} y_t(0)^2 \frac{\mathbf{1}[Z_t = 0]}{1 - p_t} \; .$$

The random variables $\widehat{A(1)}$ and $\widehat{A(0)}$ are unbiased estimates of $S(1)^2$ and $S(0)^2$ which are based on the Horvitz–Thompson principle. However, the fact that the estimators $\widehat{A(1)}$ and $\widehat{A(0)}$ are not independent and the square root is introduced means that the variance estimator $\widehat{\mathrm{VB}}$ is not an unbiased estimator for the variance bound VB. Even so, we will show that the variance estimator is consistent for the variance bound. For aesthetic considerations, we define $A(1) = S(1)^2$ and $A(0) = S(0)$ so that $\widehat{A(1)}$ is an estimator for $A(1)$ and $\widehat{A(0)}$ is an estimator for $A(0)$.

Our general approach will follow in two steps. First, by bounding its bias and variance, we will show that $\widehat{A(1)} \cdot \widehat{A(0)} - A(1)A(0)$ converges as $\tilde{\mathcal{O}}_p(T^{-1/2})$. Next, by appealing to a quantitative Continuous Mapping Theorem, we will argue that the error $\sqrt{\widehat{A(1)} \cdot \widehat{A(0)}} - \sqrt{A(1)A(0)}$ converges at the same rate. By definition, this is exactly the error of the normalized variance estimator to the normalized variance bound, i.e. $T \cdot \widehat{\mathrm{VB}} - T \cdot \mathrm{VB}$.

Before continuing, let us define new auxiliary random variables. For each $t \in [T]$, we define the variables $r_t$ and $q_t$ as

$$r_t = \frac{\mathbf{1}[z_t = 1]}{p_t} \quad \text{and} \quad q_t = \frac{\mathbf{1}[z_t = 0]}{1 - p_t} \; .$$

Below are basic facts about these auxiliary random variables.

**Lemma D.3.** *The auxiliary random variables satisfy the following properties:*

1. *$\mathbb{E}[r_t q_s] = \mathbf{1}[t \neq s]$.*

2. *$\mathbb{E}[r_t^2 \mid D_t] \leqslant \frac{1}{\delta_t}$ and $\mathbb{E}[q_t^2 \mid D_t] \leqslant \frac{1}{\delta_t}$.*

3. *The covariance $\mathrm{Cov}(r_t q_s, r_\ell q_k)$ behave in the following ways:*

$$\begin{aligned}
\mathrm{Cov}(r_t q_s, r_\ell q_k) &= 0 & \text{if } t = s \text{ or } \ell = k \\
\mathrm{Cov}(r_t q_s, r_\ell q_k) &= 0 & \text{if } t \neq s \text{ and } \ell \neq k \text{ and } \{t, s\} \cap \{\ell, k\} = \varnothing \\
\mathrm{Cov}(r_t q_s, r_\ell q_k) &= -1 & \text{if } t \neq s \text{ and } \ell \neq k \text{ and } (\, t = k \text{ or } s = \ell \,) \\
\mathrm{Cov}(r_t q_s, r_\ell q_k) &\leqslant \frac{1}{\delta_t} - 1 & \text{if } t \neq s \text{ and } \ell \neq k \text{ and } t = \ell \text{ and } s \neq k \\
\mathrm{Cov}(r_t q_s, r_\ell q_k) &\leqslant \frac{1}{\delta_s} - 1 & \text{if } t \neq s \text{ and } \ell \neq k \text{ and } t \neq \ell \text{ and } s = k \\
\mathrm{Cov}(r_t q_s, r_\ell q_k) &\leqslant \frac{1}{\delta_t \delta_s} - 1 & \text{if } t \neq s \text{ and } \ell \neq k \text{ and } t = \ell \text{ and } s = k
\end{aligned}$$

*Proof.* First, we show that $\mathbb{E}[r_t q_s] = \mathbf{1}[t \neq s]$. Let $t, s \in [T]$ and suppose that $t \neq s$. Without loss of generality, suppose that $t > s$. Then by using iterated expectation, we have that

$$\mathbb{E}[r_t q_s] = \mathbb{E}[q_s \mathbb{E}[r_t \mid D_t]] = \mathbb{E}[q_s] = 1 \; .$$

Otherwise, if $t = s$, then $r_t q_t = (\mathbf{1}[z_t = 1]/p_t) \cdot (\mathbf{1}[z_s = 0]/(1 - p_t)) = 0$ so that $\mathbb{E}[r_t q_t] = 0$.

Next, we show that $\mathbb{E}[r_t^2 \mid D_t] \leqslant \frac{1}{\delta_t}$ and $\mathbb{E}[q_t^2 \mid D_t] \leqslant \frac{1}{\delta_t}$. Observe that $\mathbb{E}[r_t^2 \mid D_t] = p_t(1/p_t^2) = 1/p_t \leqslant 1/\delta_t$, where the inequality follows by definition of Algorithm 1. A similar argument shows that $\mathbb{E}[q_t^2 \mid D_t] \leqslant 1/\delta_t$.

Finally, we establish the covariance terms one by one. We do this in order of the cases that they were presented in.

**Case 1** $(t = s)$ **or** $(\ell = k)$: If $t = s$, then $r_t q_t$ is almost surely zero, as argued above. Likewise, if $\ell = k$ then $r_\ell q_\ell$ is almost surely zero. In either of these cases, we have that $\mathrm{Cov}(r_t q_s, r_\ell q_k) = 0$.

**Case 2** $(t \neq s)$ **and** $(\ell \neq k)$ **and** $\{t, s\} \cap \{\ell, k\} = \varnothing$: Note that in this case, all the indices $t$, $s$, $\ell$, and $k$ are distinct. Without loss of generality, suppose that $t < s < \ell < k$. A repeated use of the iterated expectation yields that

$$\mathbb{E}[r_t q_s r_\ell q_s] = \mathbb{E}[r_t q_s r_\ell \mathbb{E}[q_s \mid D_s]] = \mathbb{E}[r_t q_s r_\ell] = \mathbb{E}[r_t q_s \mathbb{E}[r_\ell \mid D_r]] = \mathbb{E}[r_t q_s] = \ldots = 1 \; .$$

Because all terms of distinct, we have that $\mathbb{E}[r_t q_s] = 1$ and $\mathbb{E}[r_\ell q_k] = 1$. Thus, the covariance is equal to

$$\mathrm{Cov}(r_t q_s, r_\ell q_k) = \mathbb{E}[r_t q_s, r_\ell q_k] - \mathbb{E}[r_t q_s] \cdot \mathbb{E}[r_\ell q_k] = 1 - 1 = 0 \; .$$

**Case 3** $(t \neq s)$ **and** $(\ell \neq k)$ **and** $(t = k$ **or** $s = \ell)$: Suppose that $t = k$. In this case, observe that $r_t q_k$ is zero almost surely. Thus, $\mathbb{E}[r_t q_s r_\ell q_k] = 0$. On the other hand, $t \neq s$ and $\ell \neq k$ so that $\mathbb{E}[r_t q_s] = \mathbb{E}[r_\ell q_k] = 1$. This means that

$$\mathrm{Cov}(r_t q_s, r_\ell q_k) = \mathbb{E}[r_t q_s r_\ell q_k] - \mathbb{E}[r_t q_s] \cdot \mathbb{E}[r_\ell q_k] = 0 - 1 \cdot 1 = -1 \ .$$

The same argument shows that $s = \ell$ yields the same result.

**Case 4** $(t \neq s)$ **and** $(\ell \neq k)$ **and** $t = \ell$ **and** $s \neq k$): We begin by computing the expectation of the product of these four terms. In this case, we have that $t = \ell$ so that

$$\mathbb{E}[r_t q_s r_\ell q_k] = \mathbb{E}[r_t^2 q_s q_k] \ .$$

By assumption, the indices $t$, $s$, and $k$ are all distinct. Our approach will be to obtain an upper bound on the expectation of the project of these three terms by iterated expectation. In particular, the inequality we will use is that $\mathbb{E}[r_t^2 \mid D_t] \leqslant 1/\delta_t$. Suppose for now that $s < k < t$. In this case, we use iterated expectation to get

$$\mathbb{E}[r_t q_s r_\ell q_k] = \mathbb{E}[q_s q_k \, \mathbb{E}[r_t \mid D_t]] \leqslant \frac{1}{\delta_t} \mathbb{E}[q_s q_k] = \frac{1}{\delta_t} \mathbb{E}[q_s \, \mathbb{E}[q_k \mid D_k]] = \frac{1}{\delta_t} \mathbb{E}[q_s] = \frac{1}{\delta_t} \ .$$

In the above, we have assumes that $s < k < t$, but the same iterated expectation technique can be applied regardless of the ordering of these indices, because they are unique. Thus, in this case, $\mathbb{E}[r_t q_s r_\ell q_k] \leqslant 1/\delta_t$. Because $t \neq s$ and $\ell \neq k$, we have that $\mathbb{E}[r_t q_s] = \mathbb{E}[r_\ell q_k] = 1$. This means that

$$\mathrm{Cov}(r_t q_s, r_\ell q_k) = \mathbb{E}[r_t q_s r_\ell q_k] - \mathbb{E}[r_t q_s] \cdot \mathbb{E}[r_\ell q_k] \leqslant \frac{1}{\delta_t} - 1 \ .$$

**Case 5** $(t \neq s)$ **and** $(\ell \neq k)$ **and** $t \neq \ell$ **and** $s = k$): We begin by computing the expectation of the product of these four terms. In this case, we have that $s = k$ so that

$$\mathbb{E}[r_t q_s r_\ell q_k] = \mathbb{E}[r_t r_\ell q_s^2] \ .$$

By assumption, the indices $t$, $\ell$, and $s$ are all distinct. Using a similar argument as the previous case, we can use iterated expectation together with the bound $\mathbb{E}[q_s^2 \mid D_s] \leqslant 1/\delta_s^2$ to obtain that $\mathbb{E}[r_t q_s r_\ell q_k] \leqslant 1/\delta_s^2$. Because $t \neq s$ and $\ell \neq k$, we have that $\mathbb{E}[r_t q_s] = \mathbb{E}[r_\ell q_k] = 1$. This means that

$$\mathrm{Cov}(r_t q_s, r_\ell q_k) = \mathbb{E}[r_t q_s r_\ell q_k] - \mathbb{E}[r_t q_s] \cdot \mathbb{E}[r_\ell q_k] \leqslant \frac{1}{\delta_s} - 1 \ .$$

**Case 6** $(t \neq s)$ **and** $(\ell \neq k)$ **and** $t = \ell$ **and** $s = k$): Suppose without loss of generality that $t > s$. In this case, we can bound the product of the expectation of these four terms using iterated expectation and the proven inequalities:

$$\mathbb{E}[r_t q_s r_\ell q_k] = \mathbb{E}[r_t^2 q_s^2] = \mathbb{E}[q_s^2 \, \mathbb{E}[r_t^2 \mid D_t]] \leqslant \frac{1}{\delta_t} \mathbb{E}[q_s^2] \leqslant \frac{1}{\delta_t \delta_s} \ .$$

Because $t \neq s$, we have that $\mathbb{E}[r_t q_s] = \mathbb{E}[r_\ell, q_k] = 1$. Thus, the covariance is bounded by

$$\mathrm{Cov}(r_t q_s, r_\ell q_k) = \mathbb{E}[r_t q_s r_\ell q_k] - \mathbb{E}[r_t q_s] \cdot \mathbb{E}[r_\ell q_k] \leqslant \frac{1}{\delta_t \delta_s} - 1 \ . \qquad \square$$

First, we show that that the difference between the expected value of $\widehat{A(1)A(0)}$ and the target $A(1)A(0)$ is decreasing at a linear rate in $T$.

**Proposition D.1.** *The absolute bias of the estimated crossing term* $\widehat{A(1)A(0)}$ *to its target value* $A(1)A(0)$ *is at most*

$$\left| \mathbb{E}\big[\widehat{A(1)A(0)}\big] - S(1)^2 S(0)^2 \right| \leqslant \frac{C^4}{T} \ .$$

*Proof.* Using Lemma D.3, we can calculate the expectation of the product $\widehat{A(1)A(0)}$ as

$$\mathbb{E}\big[\widehat{A(1)A(0)}\big] = \mathbb{E}\left[ \left( \frac{1}{T} \sum_{t=1}^{T} y_t(1)^2 r_t \right) \left( \frac{1}{T} \sum_{s=1}^{T} y_s(0)^2 q_s \right) \right]$$

$$= \frac{1}{T^2} \sum_{t=1}^{T} \sum_{s=1}^{T} y_t(1)^2 y_t(0)^2 \, \mathbb{E}\big[r_t q_s\big]$$

$$= \frac{1}{T^2} \sum_{t=1}^{T} \sum_{s=1}^{T} y_t(1)^2 y_t(0)^2 - \frac{1}{T^2} \sum_{t=1}^{T} y_t(1)^2 y_t(0)^2$$

$$= \left( \frac{1}{T} \sum_{t=1}^{T} y_t(1)^2 \right) \left( \frac{1}{T} \sum_{s=1}^{T} y_s(0)^2 \right) - \frac{1}{T^2} \sum_{t=1}^{T} y_t(1)^2 y_t(0)^2$$

$$= A(1)A(0) - \frac{1}{T^2} \sum_{t=1}^{T} y_t(1)^2 y_t(0)^2 \ .$$

We complete the proof by using Cauchy-Schwarz and Assumption 1, to bound the absolute bias as

$$\left| \mathbb{E}\big[\widehat{A(1)A(0)}\big] - A(1)A(0) \right| = \frac{1}{T^2} \sum_{t=1}^{T} y_t(1)^2 y_t(0)^2$$

$$\leqslant \frac{1}{T^2} \left[ \left( \sum_{t=1}^{T} y_t(1)^4 \right) \cdot \left( \sum_{t=1}^{T} y_t(0)^4 \right) \right]^{1/2}$$

$$= \frac{1}{T} \left[ \left( \frac{1}{T} \sum_{t=1}^{T} y_t(1)^4 \right)^{1/4} \cdot \left( \frac{1}{T} \sum_{t=1}^{T} y_t(0)^4 \right)^{1/4} \right]^2$$

$$\leqslant \frac{C^4}{T} \ . \qquad \qquad \square$$

Next, we show that the variance of $\widehat{A(1)A(0)}$ is going to zero at a sufficiently fast rate.

**Proposition D.2.** *The variance of* $\widehat{A(1)A(0)}$ *is bounded as*

$$\mathrm{Var}(\widehat{A(1)A(0)}) \leqslant \frac{4 e^{\sqrt{\log(T^{1/5})}} C^8}{T} + \frac{4 C^8 e^{2\sqrt{\log(T^{1/5})}}}{T^2} = \tilde{\mathcal{O}}\Big( \frac{1}{T} \Big) \ .$$

*Proof.* We begin by decomposing the variance of $\widehat{A(1)A(0)}$ into covariances of products of the auxiliary random variables $r_t$ and $q_s$. To this end, observe that

$$\mathrm{Var}(\widehat{A(1)A(0)}) = \mathrm{Var}\Big( \frac{1}{T^2} \sum_{t=1}^{T} \sum_{s=1}^{T} y_t(1)^2 y_t(0)^2 r_t q_s \Big)$$

$$= \frac{1}{T^4} \sum_{t=1}^{T} \sum_{s=1}^{T} \sum_{\ell=1}^{T} \sum_{k=1}^{T} y_t(1)^2 y_s(0)^2 y_\ell(1)^2 y_k(0)^2 \, \mathrm{Cov}(r_t q_s, r_\ell q_k)$$

Next, we will use the result of Lemma D.3 to handle the individual covariance terms. In particular, the first six types of terms, as described in Lemma D.3. The first three types of terms are at most 0, so we may discard them from the sum, as they contribute no positive value. The last three terms have upper bounds, which we use here to obtain the following upper bound:

$$\leqslant \underbrace{\frac{1}{T^4} \sum_{t=1}^{T} \sum_{s\in[T]\setminus\{t\}} \sum_{k\in[T]\setminus\{t,s\}} y_t(1)^4 y_s(0)^2 y_k(0)^2 \Big( \frac{1}{\delta_t} - 1 \Big)}_{\text{Term } T_1}$$

$$+ \underbrace{\frac{1}{T^4} \sum_{t=1}^{T} \sum_{s\in[T]\setminus\{t\}} \sum_{\ell\in[T]\setminus\{t,s\}} y_t(1)^2 y_\ell(1)^2 y_s(0)^4 \Big( \frac{1}{\delta_s} - 1 \Big)}_{\text{Term } T_2}$$

$$+ \underbrace{\frac{1}{T^4} \sum_{t=1}^{T} \sum_{s\neq t} y_t(1)^4 y_s(0)^4 \Big( \frac{1}{\delta_t \delta_s} - 1 \Big)}_{\text{Term } T_3}$$

Our goal will now be to bound each of these terms individually.

**Terms 1 and 2**: Terms 1 and 2 are similar and will be handled in the same way. Let's begin with Term 1. Observe that by Lyapunov's inequality, the moment assumptions, and Lemma D.2, we have that

$$
\begin{aligned}
T_1 &\leqslant \frac{1}{\delta_T \cdot T^4} \Big( \sum_{t=1}^{T} y_t(1)^4 \Big) \Big( \sum_{t=1}^{T} y_t(0)^2 \Big)^2 \\
&\leqslant \frac{2e^{\sqrt{\log(T^{1/5})}}}{T} \Big( \frac{1}{T} \sum_{t=1}^{T} y_t(1)^4 \Big) \Big( \frac{1}{T} \sum_{t=1}^{T} y_t(0)^2 \Big)^2 \qquad\qquad \text{(Lemma D.2)} \\
&= \frac{2e^{\sqrt{\log(T^{1/5})}}}{T} \Big[ \Big( \frac{1}{T} \sum_{t=1}^{T} y_t(1)^4 \Big)^{1/4} \Big( \frac{1}{T} \sum_{t=1}^{T} y_t(0)^2 \Big)^{1/2} \Big]^4 \\
&\leqslant \frac{2e^{\sqrt{\log(T^{1/5})}}}{T} \Big[ \Big( \frac{1}{T} \sum_{t=1}^{T} y_t(1)^4 \Big)^{1/4} \Big( \frac{1}{T} \sum_{t=1}^{T} y_t(0)^4 \Big)^{1/4} \Big]^4 \qquad \text{(Lyapunov's inequality)} \\
&\leqslant \frac{2e^{\sqrt{\log(T^{1/5})}}C^8}{T} \qquad\qquad\qquad\qquad\qquad\qquad\qquad\qquad \text{(Assumption 1)}
\end{aligned}
$$

A similar argument shows that $T_2 \leqslant (2e^{\sqrt{\log(T^{1/5})}}C^8)/T$.

**Term 3**: The third term may be upper bounded using Lemma D.2 and the moment assumptions. Namely,

$$
\begin{aligned}
T_3 &\leqslant \frac{1}{\delta_T^2 T^4} \Big( \sum_{t=1}^{T} y_t(1)^4 \Big) \Big( \sum_{t=1}^{T} y_t(0)^4 \Big) \\
&= \frac{\big(2e^{\sqrt{\log(T^{1/5})}}\big)^2}{T^2} \Big( \frac{1}{T} \sum_{t=1}^{T} y_t(1)^4 \Big) \Big( \frac{1}{T} \sum_{t=1}^{T} y_t(0)^4 \Big) \qquad \text{(Lemma D.2)} \\
&\leqslant \frac{4C^8 e^{2\sqrt{\log(T^{1/5})}}}{T^2} \qquad\qquad\qquad\qquad\qquad\qquad\qquad\qquad \text{(Assumption 1)}
\end{aligned}
$$

Taken together, this shows that

$$
\mathrm{Var}(\widehat{A(1)A(0)}) \leqslant \frac{4e^{\sqrt{\log(T^{1/5})}}C^8}{T} + \frac{4C^8 e^{2\sqrt{\log(T^{1/5})}}}{T^2} = \tilde{\mathcal{O}}\Big( \frac{1}{T} \Big) \ .
$$

$\square$

This establishes the following corollary, which shows that the error $\widehat{A(1)A(0)} - A(1)A(0)$ is going to zero at a near-parametric rate, which follows from by Chebyshev's inequality from Propositions D.1 and D.2.

**Corollary D.1.** *The following error goes to zero:* $\widehat{A(1)A(0)} - A(1)A(0) = \tilde{\mathcal{O}}_p\big(T^{-1/2}\big).$

Using the results derived above, we are ready to prove Theorem 5.1, which we restate here for convenience.

**Theorem 5.1.** *Under Assumptions 1 and 2, and the parameters stated in Theorem 4.2, the error of the normalized variance estimator under* CLIP-OGD *is* $T \cdot \widehat{VB} - T \cdot VB = \tilde{\mathcal{O}}_p(T^{-1/2}).$

*Proof of Theorem 5.1.* Recall that the variance estimator and the variance bound are equal to $T \cdot VB = 4S(1)S(0) = 4\sqrt{A(1)A(0)}$ and $T \cdot \widehat{VB} = 4\sqrt{\widehat{A(1)A(0)}}$ so that the error is given by

$$
T \cdot \widehat{VB} - T \cdot VB = 4\Big[ \sqrt{A(1)A(0)} - \sqrt{\widehat{A(1)A(0)}} \Big] \ .
$$

Corollary D.1 states that the error $\widehat{A(1)A(0)} - A(1)A(0)$ is on the order of $\tilde{\mathcal{O}}_p\big(T^{-1/2}\big)$. By Assumption 1, we have that $A(1)A(0) = \sqrt{S(1)^2 S(0)^2} > c^2$. Observe that the square root function $g(x) = \sqrt{x}$ is Lipschitz on the interval $(c^2, \infty)$. Thus, by a rate-preserving Continuous Mapping Theorem, we have that the error of the normalized variance estimator is on the same order i.e. $T \cdot VB - T \cdot \widehat{VB} = \sqrt{A(1)A(0)} - \sqrt{\widehat{A(1)A(0)}} = \tilde{\mathcal{O}}_p\big(T^{-1/2}\big).$ $\square$

## D.2 Valid Confidence Intervals (Corollary 5.1)

We now prove the asymptotic validity of the associated Chebyshev-type intervals. This proof is standard in the design-based literature, but we present it here for completeness.

**Corollary 5.1.** *Under Assumptions 1 and 2, and parameters stated in Theorem 4.2, Chebyshev-type intervals are asymptotically valid: for all $\alpha \in (0, 1]$, $\liminf_{T\to\infty} \Pr(\tau \in \hat{\tau} \pm \alpha^{-1/2}\sqrt{\widehat{VB}}) \geqslant 1 - \alpha$.*

*Proof.* Define the random variables

$$
Z = \frac{\tau - \hat{\tau}}{\sqrt{\mathrm{Var}(\hat{\tau})}} \quad \text{and} \quad Z' = \frac{\tau - \hat{\tau}}{\sqrt{\widehat{VB}}} \ .
$$

Observe that they are related in the following way:

$$
Z' = \frac{\tau - \hat{\tau}}{\sqrt{\widehat{VB}}} = \frac{\tau - \hat{\tau}}{\sqrt{\mathrm{Var}(\hat{\tau})}} \cdot \left( \sqrt{\frac{\mathrm{Var}(\hat{\tau})}{VB}} \cdot \sqrt{\frac{VB}{\widehat{VB}}} \right) = Z \cdot \left( \sqrt{\frac{\mathrm{Var}(\hat{\tau})}{VB}} \cdot \sqrt{\frac{T \cdot VB}{T \cdot \widehat{VB}}} \right) \ .
$$

By definition, we have that $\limsup_{T\to\infty} \mathrm{Var}(\hat{\tau})/VB \leqslant 1$. Recall that by Proposition B.1, $T \cdot VB \geqslant T \cdot V_N = \Omega(1)$ so that by Theorem 5.1 and Continuous Mapping Theorem, we have that $\sqrt{\frac{T \cdot VB}{T \cdot \widehat{VB}}} \xrightarrow{p} 1$. Thus, by Slutsky's theorem we have that $Z'$ is asymptotically stochastically dominated by $Z$. Now we are ready to compute the coverage probability.

$$
\begin{aligned}
\liminf_{T\to\infty} \Pr\left(\tau \in \hat{\tau} \pm \alpha^{-1/2}\sqrt{\widehat{VB}}\right) &= \liminf_{T\to\infty} \Pr\left(\left|\frac{\tau - \hat{\tau}}{\sqrt{\widehat{VB}}}\right| \leqslant \alpha^{-1/2}\right) \\
&= \liminf_{T\to\infty} \Pr\left(|Z'| \leqslant \alpha^{-1/2}\right) \\
&\geqslant \liminf_{T\to\infty} \Pr\left(|Z| \leqslant \alpha^{-1/2}\right) \\
&\geqslant 1 - \alpha \ ,
\end{aligned}
$$

where the last line followed from Chebyshev's inequality and the fact that $\mathrm{Var}(Z) = 1$. $\qquad\square$

# E  Analysis of Alternative Designs

In this section, we provide analysis on the efficacy of existing adaptive experimental designs for the problem of Adaptive Neyman Allocation. To this end, we show two negative results. In Section E.1, we show that the two-stage design of [Hahn et al., 2011, Blackwell et al., 2022] (i.e. Explore–then–Commit) can suffer linear expected Neyman Regret in the design-based framework for a large class of potential outcome sequences. In Section E.2, we show that mutli-arm bandit algorithms which achieve sublinear expected outcome regret will incur super-linear expected Neyman regret, providing further evidence that these two goals are incompatible.

## E.1  Analysis of Explore-then-Commit (Proposition 6.1)

In this setting, we show that Explore-then-Commit designs can sometimes suffer linear Neyman regret, and therefore not recover the Neyman variance in large samples. We formally introduce our definition of Explore-then-Commit designs below. Let $p^*_{T_0}$ be defined as

$$
p^*_{T_0} = \left( 1 + \sqrt{\frac{\sum_{t=1}^{T_0} y_t(0)^2}{\sum_{t=1}^{T_0} y_t(1)^2}} \right)^{-1} ,
$$

which is the optimal Neyman probability when considering only the sample up to $T_0$.

Our definition of ETC encompasses many possible ways of estimating the optimal treatment probability. The only requirement is that the estimation method will converge to $p^*_{T_0}$ at the rate $T_0^{-1/2}$. We consider $p^*_{T_0}$ rather than the true Neyman probability $p^*$ because the observed data is informative only of the outcomes in the exploration phase $T_0$. Many natural estimators will fall into this class, including Horvitz–Thompson style estimators similar to those used in the construction of our variance estimator.

**Algorithm 2:** EXPLORE-THEN-COMMIT

---

**Input:** Exloration phase size $T_0$
// Explore Phase
**for** $t = 1 \ldots T_0$ **do**
$\quad$ Sample treatment assignment $Z_t$ as 1 with probability $1/2$ and 0 with probability $1/2$
$\quad$ Observe outcome $Y_t = \mathbf{1}[Z_t = 1]y_t(1) + \mathbf{1}[Z_t = 0]y_t(0)$
**end**
Construct an estimator $\widehat{p^*_{T_0}}$ from observed data $(Z_1, Y_1, \ldots Z_{T_0}, Y_{T_0})$ satisfying $\widehat{p^*_{T_0}} - p^*_{T_0} = \mathcal{O}_p(T_0^{-1/2})$.
// Exploit Phase
**for** $t = T_0 + 1 \ldots T$ **do**
$\quad$ Sample treatment assignment $Z_t$ as 1 with probability $\widehat{p^*_{T_0}}$ and 0 with probability $1 - \widehat{p^*_{T_0}}$
$\quad$ Observe outcome $Y_t = \mathbf{1}[Z_t = 1]y_t(1) + \mathbf{1}[Z_t = 0]y_t(0)$
**end**

---

Before continuing, we provide a few more definitions. We define the second moments of treatment and control outcomes as well as the correlation in the exploration phase as

$$S_{T_0}(1)^2 = \frac{1}{T}\sum_{t=1}^{T_0} y_t(1)^2 \quad S_{T_0}(0)^2 = \frac{1}{T}\sum_{t=1}^{T_0} y_t(0)^2 \quad \text{and} \quad \rho_{T_0} = \frac{\frac{1}{T_0}\sum_{t=1}^{T} y_t(1)y_t(0)}{S_{T_0}(1)S_{T_0}(0)} \quad.$$

We are now ready to state the formal version of Proposition 6.1.

**Proposition 6.1\*.** *Suppose that $T_0 = \Omega(T^\epsilon)$ for some $\epsilon > 0$ and further suppose that the outcome sequence satisfies the following properties for constants $C \geqslant c > 0$ and $c' > 0$:*

- *The second moments $S(1)$, $S(0)$, $S_{T_0}(1)$, and $S_{T_1}(0)$ are contained in the interval $[c, C]$.*

- *The correlations are bounded away from -1, i.e. $\rho_{T_0}, \rho \geqslant -1 + c$.*

- *The second moments satisfy the following:*

$$S(1)^2\Big(\frac{S(0)}{S(1)} - \frac{S_{T_0}(0)}{S_{T_0}(1)}\Big) + S(0)^2\Big(\frac{S(1)}{S(0)} - \frac{S_{T_0}(1)}{S_{T_0}(0)}\Big) \geqslant c'$$

*Then, the Neyman Regret of Explore-then-Commit is at least linear in probability, $\mathcal{R}_T = \Omega_p(T)$.*

The first two conditions are essentially extensions of Assumption 1 to the exploration phase. This ensures that the probability $p^*_{T_0}$ (which is estimated in the Explore-then-Commit design) does not approach 0 or 1. The third condition is what really makes Explore-then-Commit fail to achieve sublinear Neyman regret. This condition states that the ratio of the second moments in the exploration phase is different than in the larger sequence. For example, if $S(1) = S(0) = 1$ but $S_{T_0}(0)/S_{T_0}(1) = 2$ then the condition would hold. In this case, we should not expect Explore-then-Commit to achieve the Neyman variance because the exploration phase does not contain sufficient information about the optimal Neyman probability. We now prove the proposition.

*Proof.* Let $p^* = \arg\min_{p\in[0,1]} \sum_{t=1}^{T} f_t(p)$ be the Neyman probability. We begin by re-arranging terms in the Neyman regret:

$$\mathcal{R}_T = \sum_{t=1}^{T} f_t(p_t) - \sum_{t=1}^{T} f_t(p^*) \qquad\qquad \text{(def of Neyman regret)}$$

$$= \sum_{t=1}^{T_0} f_t(p_t) - f_t(p^*) + \sum_{t=T_0+1}^{T} f_t(p_t) - f_t(p^*) \qquad\qquad \text{(splitting terms by phases)}$$

$$= \sum_{t=1}^{T_0} f_t(1/2) - f_t(p^*) + \sum_{t=T_0+1}^{T} f_t(\widehat{p^*_{T_0}}) - f_t(p^*) \qquad\qquad \text{(def of ETC)}$$

$$= \sum_{t=1}^{T_0} f_t(1/2) - f_t(\widehat{p^*_{T_0}}) + \sum_{t=1}^{T} f_t(\widehat{p^*_{T_0}}) - f_t(p^*) \qquad\qquad \text{(adding + subtracting)}$$

$$\geqslant \sum_{t=1}^{T_0} f_t(p_{T_0}^*) - f_t(\widehat{p_{T_0}^*}) + \sum_{t=1}^{T} f_t(\widehat{p_{T_0}^*}) - f_t(p^*) \ ,$$

Where the inequality follows by the optimality of $p_{T_0}^*$ on the exploration phase. By adding and subtracting the $f_t(p_{T_0}^*)$ to the second sum, we obtain the following decomposition

$$= \underbrace{\sum_{t=1}^{T_0} f_t(p_{T_0}^*) - f_t(\widehat{p_{T_0}^*})}_{\text{Term 1}} + \underbrace{\sum_{t=1}^{T} f_t(\widehat{p_{T_0}^*}) - f_t(p_{T_0}^*)}_{\text{Term 2}} + \underbrace{\sum_{t=1}^{T} f_t(p_{T_0}^*) - f_t(p^*)}_{\text{Term 3}}$$

We handle each of the terms separately in the remainder of the proof.

**Term 1**: By the assumptions on the second moments and correlation of outcomes in the exploration phase, we have that $p_{T_0}^*$ is bounded away from 0 and 1 by a constant. Furthermore, the cost functions $f_t$ are Lipschitz on the interval $[\gamma, 1 - \gamma]$ for any fixed $\gamma$. Thus, we may apply the quantitative Continuous Mapping Theorem to bound the absolute value of Term 1 as follows:

$$|\sum_{t=1}^{T_0} f_t(p_{T_0}^*) - f_t(\widehat{p_{T_0}^*})| \leqslant \sum_{t=1}^{T_0} |f_t(p_{T_0}^*) - f_t(\widehat{p_{T_0}^*})| \qquad \text{(triangle inequality)}$$

$$\leqslant \sum_{t=1}^{T_0} \mathcal{O}_p(T_0^{-1/2}) \qquad \text{(estimator property + CMT)}$$

$$= \mathcal{O}_p(T_0 \cdot T_0^{-1/2})$$

$$= \mathcal{O}_p(T_0^{1/2})$$

$$= \mathcal{O}_p(T^{1/2}) \ ,$$

where the final inequality follows from the fact that $T_0 \leqslant T$.

**Term 2**: A similar argument may be applied to the second term. Again, we may apply the continuous mapping theorem as before to obtain

$$|\sum_{t=1}^{T} f_t(p_{T_0}^*) - f_t(\widehat{p_{T_0}^*})| \leqslant \sum_{t=1}^{T} |f_t(p_{T_0}^*) - f_t(\widehat{p_{T_0}^*})| \qquad \text{(triangle inequality)}$$

$$\leqslant \sum_{t=1}^{T} \mathcal{O}_p(T_0^{-1/2}) \qquad \text{(estimator property + CMT)}$$

$$= \mathcal{O}_p(T \cdot T_0^{-1/2})$$

$$= \mathcal{O}_p(T \cdot T^{-\epsilon/2})$$

$$= \mathcal{O}_p(T^{1-\epsilon/2}) \ ,$$

where we have used the assumption that $T_0 = \Omega(T^\epsilon)$.

**Term 3**: We now handle the third term. Let $V_0$ be the variance of adaptive Horvitz–Thompson estimator under the Bernoulli design when using the treatment probability $p_{T_0}^*$. By construction of the cost functions, we have that they are related to the variance as follows:

$$\sum_{t=1}^{T} f_t(p_{T_0}^*) - f_t(p^*)$$

$$= T \cdot \left[ T \cdot V_0 - T \cdot V_N \right]$$

$$= T \cdot \left[ \left( S(1)^2 \left\{ \frac{1}{p_{T_0}^*} - 1 \right\} + S(0)^2 \left\{ \frac{1}{1 - p_{T_0}^*} - 1 \right\} \right) - \left( S(1)^2 \left\{ \frac{1}{p^*} - 1 \right\} + S(0)^2 \left\{ \frac{1}{1 - p^*} - 1 \right\} \right) \right]$$

$$= T \cdot \left[ S(1)^2 \left( \frac{1}{p_{T_0}^*} - \frac{1}{p^*} \right) + S(0)^2 \left( \frac{1}{1 - p_{T_0}^*} - \frac{1}{1 - p^*} \right) \right]$$

$$= T \cdot \left[ S(1)^2 \Big( \frac{S(0)}{S(1)} - \frac{S_{T_0}(0)}{S_{T_0}(1)} \Big) + S(0)^2 \Big( \frac{S(1)}{S(0)} - \frac{S_{T_0}(1)}{S_{T_0}(0)} \Big) \right] \ ,$$

where the last equality follows by definition of the probabilities. By Assumption, we have that this bracketed term is constant so that the third term is linear in $T$.

Putting these together, we have that the Neyman regret is lower bounded as

$$\mathcal{R}_T \geqslant \Omega(T) - \mathcal{O}_p(T^{1-\epsilon/2}) - \mathcal{O}_p(T^{1/2}) = \Omega_p(T) \ . \qquad \square$$

### E.2  Analysis of Designs for Outcome Regret (Proposition 6.2)

In this section, we prove Proposition 6.2, which establishes that outcome regret and Neyman regret cannot be simultaneously minimized in general. We restate a more formal version of the proposition here. In order for a simpler proof, we make restrictions that the units have constant treatment effect and that each of the individual outcomes are more strictly bounded. We conjecture that the trade-off between Neyman and outcome regret will hold under weaker conditions.

**Proposition 6.2\*.** *Let $\mathcal{A}$ be an adaptive treatment algorithm achieving sublinear outcome regret, i.e. there exists $q \in (0,1)$ such that $\mathbb{E}[\mathcal{R}_T^{outcome}] \leqslant O(T^q)$ for all outcome sequences satisfying Assumption 1. Consider an outcome sequence satisfying Assumption 1 with constants $C \geqslant c > 0$ and the additional conditions:*

- *$\max_{1 \leqslant t \leqslant T} y_t(0)^2 \leqslant C^2$*

- *For all $t \in [T]$, $y_t(1) - y_t(0) = \tau$ and $\tau > c'$ for a constant $c' > 0$.*

*Then, $\mathcal{A}$ suffers super-linear Neyman regret on this outcome sequence: $\mathbb{E}[\mathcal{R}_T] \geqslant \Omega(T^{2-q})$.*

*Proof.* To begin, we re-express the outcome regret in terms of the expected treatment probabilities played by algorithm $\mathcal{A}$. Observe that the expected outcome regret may be written as

$$\mathbb{E}[\mathcal{R}_T^{\text{outcome}}] = \mathbb{E}\left[ \max_{k \in \{0,1\}} \sum_{t=1}^{T} y_t(k) - \sum_{t=1}^{T} Y_t \right] \qquad \text{(def of regret)}$$

$$= \sum_{t=1}^{T} y_t(1) - \sum_{t=1}^{T} \mathbb{E}[Y_t] \qquad (\tau > 0)$$

$$= \sum_{t=1}^{T} y_t(1) - \sum_{t=1}^{T} \mathbb{E}\big[ y_t(1)\mathbf{1}[Z_t = 1] + y_t(0)\mathbf{1}[Z_t = 0] \big]$$

$$= \sum_{t=1}^{T} y_t(1) - \sum_{t=1}^{T} y_t(1)\, \mathbb{E}[p_t] + y_t(0) \cdot \big(1 - \mathbb{E}[p_t]\big)$$

$$= \sum_{t=1}^{T} \big(y_t(1) - y_t(0)\big) \cdot \big(1 - \mathbb{E}[p_t]\big)$$

$$= \tau \cdot \sum_{t=1}^{T} \big(1 - \mathbb{E}[p_t]\big) \ ,$$

where the last equality follow as $y_t(1) - y_t(0) = \tau$ for all $t \in [T]$ by assumption. Because the outcome sequence satisfies Assumption 1, the expected outcome regret is at most $\mathbb{E}[\mathcal{R}_T^{\text{outcome}}] \leqslant \beta \cdot T^q$ for some constant $\beta$. By the above, this implies that the expectation of the sum of probabilities $1 - p_t$ must be small,

$$\sum_{t=1}^{T} \big(1 - \mathbb{E}[p_t]\big) \leqslant \frac{\beta}{\tau} T^q \ .$$

Next, we show that $\mathcal{A}$ must incur a large cost with respect to the functions $f_t$ in the definition of Neyman regret. To do this, we will use a weighted version of the AM-HM inequality which states that for $x_1 \ldots x_T > 0$ and $w_1 \ldots w_n \geqslant 0$, we have that

$$\frac{\sum_{t=1}^{T} w_t x_t}{\sum_{t=1}^{T} w_t} \geqslant \frac{\sum_{t=1}^{T} w_t}{\sum_{t=1}^{T} \frac{w_t}{x_t}} \ .$$

The usual AM-HM inequality is recovered when $w_t = 1/T$. We now bound the expected Neyman loss, observing that

$$
\mathbb{E}\Big[\sum_{t=1}^{T} f_t(p_t)\Big] = \mathbb{E}\Big[\sum_{t=1}^{T} \frac{y_t(1)^2}{p_t} + \frac{y_t(0)^2}{1-p_t}\Big] \qquad \text{(def of } f_t)
$$

$$
\geqslant \mathbb{E}\Big[\sum_{t=1}^{T} \frac{y_t(0)^2}{1-p_t}\Big] \qquad \text{(non-negativity)}
$$

$$
= \sum_{t=1}^{T} y_t(0)^2 \cdot \mathbb{E}\Big[\frac{1}{1-p_t}\Big] \qquad \text{(linearity of } \mathbb{E}[\cdot])
$$

$$
\geqslant \sum_{t=1}^{T} y_t(0)^2 \cdot \frac{1}{\mathbb{E}[1-p_t]} \qquad \text{(Jensen's inequality)}
$$

$$
\geqslant \frac{\left(\sum_{t=1}^{T} y_t(0)^2\right)^2}{\sum_{t=1}^{T} y_t(0)^2 \cdot \mathbb{E}[1-p_t]} \qquad \text{(weighted AM-HM)}
$$

$$
\geqslant T^2 \frac{\left(\frac{1}{T}\sum_{t=1}^{T} y_t(0)^2\right)^2}{\max_{1 \leqslant t \leqslant T} y_t(0)^2} \cdot \frac{1}{\sum_{t=1}^{T} \mathbb{E}[1-p_t]}
$$

$$
= T^2 \frac{S(0)^2}{\max_{1 \leqslant t \leqslant T} y_t(0)^2} \cdot \frac{1}{\sum_{t=1}^{T} \mathbb{E}[1-p_t]}
$$

$$
\geqslant T^2 \frac{c^2}{C^2} \cdot \frac{\tau}{\beta} T^{-q}
$$

$$
\geqslant \frac{c^2 c'}{C^2 \beta} T^{2-q} \ ,
$$

where the last two inequalities follow from moment bounds in Assumption 1 together with the assumptions on the outcome sequence stated in the Theorem.

Next, we show that the optimal Neyman design incurs a much smaller cost. In particular,

$$
\min_{p \in [0,1]} \sum_{t=1}^{T} f_t(p) \leqslant \sum_{t=1}^{T} f_t(1/2)
$$

$$
= \sum_{t=1}^{T} \frac{y_t(1)^2}{1/2} + \frac{y_t(0)^2}{1/2}
$$

$$
= 2T\left[\frac{1}{T}\sum_{t=1}^{T} y_t(1)^2 + \frac{1}{T}\sum_{t=1}^{T} y_t(0)^2\right]
$$

$$
= 2T(S(1)^2 + S(0)^2)
$$

$$
\leqslant 4C^2 T \ .
$$

Together, these facts establish that the Neyman regret for $\mathcal{A}$ is lower bounded as

$$
\mathbb{E}[\mathcal{R}_T] = \mathbb{E}\Big[\sum_{t=1}^{T} f_t(p_t) - \min_{p \in [0,1]} \sum_{t=1}^{T} f_t(p)\Big] \geqslant \frac{c^2 c'}{C^2 \beta} T^{2-q} - 4C^2 T \geqslant \Omega(T^{2-q}) \ . \qquad \square
$$

# F   Ethical Considerations

There are—at least—two objectives when constructing an adaptive treatment allocation.

- **Minimizing Cumulative Regret**: give the "best" treatment to as many people as possible.

- **Minimizing Variance of the Effect Estimate**: estimate the effect of the treatment to as high precision as possible.

As we show in Proposition 6.2, these two objective are fundamentally incompatible: an adaptive design which aims to estimate the effect to high precision must assign treatment which has worse outcomes. Likewise, a design which seeks to maximize the utility of assigned treatments will not be able to reliably estimate causal effects to high precision. Which one of these is more ethically desirable depends on the purpose and the context of the experiment. For guidance on this ethical question, we turn to The Belmont Report.

In 1979, the National Commission for the Protection of Human Subjects of Biomedical and Behavioral Research released the "Belmont Report", which has been one of the foundational texts for ethical guidance in research conducted with human subjects [for the Protection of Human Subjects of Biomedical and Research, 1978]. One of the three basic ethical principles laid out in the report is "benevolence" which is understood as an the obligation the researcher has for the research to improve the general well-being of society, including those people in the experiment. The report writes:

> The Hippocratic maxim 'do no harm' has long been a fundamental principle of medical ethics. Claude Bernard extended it to the realm of research, saying that one should not injure one person regardless of the benefits that might come to others. However, even avoiding harm requires learning what is harmful; and, in the process of obtaining this information, persons may be exposed to risk of harm. Further, the Hippocratic Oath requires physicians to benefit their patients 'according to their best judgment.' Learning what will in fact benefit may require exposing persons to risk. The problem posed by these imperatives is to decide when it is justifiable to seek certain benefits despite the risks involved, and when the benefits should be foregone because of the risks.

From this perspective, it may be ethically advisable to use a variance minimizing adaptive design because it allows the researcher to learn the effect while subjecting fewer human subjects to the experimental treatments. In other words, a variance minimizing design allows researchers to learn what is harmful and what is beneficial while subjecting fewer human subjects to possible harm. An adaptive treatment plan which minimizes cumulative regret will ensure that minimal harm is done to subjects in the experiment, but will offer less certainty about the extent of the benefit or harm of the treatments. Such an approach will lead to less informative generalizable knowledge of treatment effects, possibly defeating the goal of the research study.

That being said, it is not our goal to suggest that one adaptive allocation plan is most ethical in all circumstances. Indeed, these ethical questions have no systematic answers which are generally applicable. It is the burden of the researchers to carefully "decide when it is justifiable to seek certain benefits despite the risks involved, and when the benefits should be foregone because of the risks." Our goal in this work is merely to provide improved statistical methodology which affords the researchers more choices when addressing these ethical questions.

