# OpenReview forum: "CLIP-OGD: An Experimental Design for Adaptive Neyman Allocation in Sequential Experiments"
_NeurIPS.cc/2023/Conference — NeurIPS 2023 spotlight_

### Official Review · Reviewer_wYBx · 2023-07-02

**Soundness:** 3 good
**Presentation:** 3 good
**Contribution:** 2 fair
**Rating:** 6
**Confidence:** 2

**Summary:**

This paper studies adaptive Neymann allocation and proposed an algorithm that achieves expected Neymann regret $\tilde{O}(\sqrt{T})$. I am not an expert in sequential experiment design so my ability is limited to assess the impact/relevance of this paper. Yet, I do consider myself well-versed in the potential-outcomes framework.

**Strengths:**

- I found the writing and analyses appear to be rigorous and precise.
- The proposed algorithm achieves the Neymann variance in the synthetic experiment.

**Weaknesses:**

See "Questions" for more details.

**Questions:**

1. I understand the authors may wish to provide sufficient background, but it is a bit hard to tell what are existing (and well-known) results and what are new results (except starting from page 6 when the authors introduce the algorithm).
2. I felt Assumption 1 is a bit tricky to decipher: what is the motivation to have the same constant $c$? What are the scenarios where this assumption may break?
3. Have you considered combining CLIP-OGD with an outcome model to get a doubly-robust estimator?
4. Can you add more simulations?

---

> ### Author Rebuttal · Authors · 2023-08-08
>
> # Response to Reviewer 5 (wYBx)
>
> We thank you for your thoughtful reading of our paper.
> We are happy to hear that you found the writing and analysis to be rigorous and precise.
> Moreover, your comments have helped us improve the paper in places where there were ambiguities or confusion.
>
> We respond to the questions raised in your review below.
>
> ## Question 1 - Existing and New Results
>
> > I understand the authors may wish to provide sufficient background, but it is a bit hard to tell what are existing (and well-known) results and what are new results (except starting from page 6 when the authors introduce the algorithm).
>
> Thank you for highlighting this lack of clarity in our writing.
> Let us summarize the separation between existing work and our main contributions here.
>
> - **Existing Work**: The main existing results which are similar in spirit to ours are the works of (Hahn et al 2011) and (Blackwell et al 2022) which propose and formally analyze experimental designs for Adaptive Neyman Allocation. Neither of these works are able to show that the optimal Neyman variance is achieved, even in large samples.
>
> In this paper, our main contributions are:
>
> - (Section 4.1): The introduction of performance measures which ensure that the optimal Neyman variance is achieved in large samples.
> - (Section 4.2): A new adaptive experimental design which ensures bounds on the Neyman regret, thereby ensuring that the optimal Neyman variance is achieved.
> - (Section 5): Methods for valid confidence intervals using this experimental design.
>
> There are two additional minor contributions are contained in Section 6:
>
> - Proposition 6.1 shows that the "explore-then-commit" style designs proposed by existing work (Hahn et al 2011) and (Blackwell et al 2022) **can not attain optimal Neyman variance** in a design-based framework without very strong homogeneity assumptions.
> - Proposition 6.2 shows that MAB algorithms which minimize outcome regret are incompatible with achieving optimal Neyman variance. In fact, the variance of MAB algorithms will have convergence rates **larger by orders of magnitude**.
>
> The background in Section 2 contains the formulation of design-based inference as well as basic facts about the adaptive Horvitz--Thompson estimator.
> These basic facts have been established in i.i.d. settings but perhaps not design-based settings, although this does not count as a contribution by any means and we do not mean for it to be interpreted as such.
> We will add a citation to the i.i.d. literature to make this clear.
>
> In order to address your question, we will revise the introduction, related work, and preliminaries sections to clarify these points.
>
> ## Question 2 -- Assumption 1
>
> > I felt Assumption 1 is a bit tricky to decipher: what is the motivation to have the same constant $c$
> ? What are the scenarios where this assumption may break?
>
> We appreciate your comment here, which highlights that Assumption 1 would benefit from a longer discussion which more carefully interprets its scope conditions.
> To address your comment, we will include in the paper a discussion similar to the one below (although more streamlined):
>
> As stated briefly at the end of Section 2, the purpose of Assumption 1 is to ensure that the optimal Neyman variance is bounded away from zero.
> More precisely, the Assumption guarantees that $V_N = \Omega (1 / T)$ or on a normalized scale, $T \cdot V_N = \Omega(1)$.
> This is important because without this assumption, it is impossible to construct a sequential experimental design that can achieve a comparable variance.
>
> To illustrate this, consider the following example where Assumption 1 does not hold: for all units $i \in [n]$, consider the outcomes
>
> $$
> y_t(1) = 1
> \quad \text{and} \quad
> y_t(0) = -1
> \enspace.
> $$
>
> Here, $\rho = -1$ so that the outcomes are perfectly negatively correlated and thus Assumption 1 does not hold.
> Note that the average treatment effect is $\tau = 2$.
> The optimal Neyman design is to set $p^* = 1/2$.
> In this case, the Horvitz--Thompson estimator is $\hat{\tau} = 2$ with probability 1 and thus the Neyman variance is zero, i.e. $V_N = 0$.
> Thus, any sequential design which can select treatment probability $P_t \neq 1/2$ at some time $t$ will have a positive variance, and thus unbounded Neyman ratio.
>
> The construction above requires a somewhat pathological set of outcomes -- even a slight perturbation of the outcomes is enough to prevent such pathologies.
>  We remark that Assumption 1 (or similar assumptions) are typically required for efficiently estimating variance and constructing confidence intervals.
> For these reasons, experimenters are typically comfortable making assumptions like Assumption 1, which rule out pathological edge cases.
>
> The fact that the same constant $c$ is used in both parts of the assumption is arbitrary: it can be replaced by two separate constants $c_1$ and $c_2$ and our analysis goes through by taking $c = \min \{ c_1, c_2 \}$.
>
> ## Question 3 --  Doubly Robust Estimation
>
> > Have you considered combining CLIP-OGD with an outcome model to get a doubly-robust estimator?
>
> This is an excellent question!
> We have not considered such extensions, but they sound very interesting.
> We are not aware of doubly robust estimation in an adaptive design.
> A promising direction for future work!
>
> ## Question 4 -- More Simulations
>
> > Can you add more simulations?
>
> Thank you for encouraging us to increase the strength of our simulations.
> Please see "Global Response" for our discussion on the results of a MAB baseline and additional new simulation results.

---

> > ### Comment · Reviewer_wYBx · 2023-08-16
> > **Thank you for the reply**
> >
> > I thank the authors for the detailed response and addressing my concerns. I agree that with a more streamlined and detailed discussions on Assumption 1, and with the new results this paper is stronger that it was.

---

> > > ### Author Response · Authors · 2023-08-17
> > > **response to reply**
> > >
> > > Thank you for your response. We are happy to hear that you found that the revisions brought forth in the rebuttals increased the strength of the paper.

---

### Official Review · Reviewer_etbf · 2023-07-06

**Soundness:** 3 good
**Presentation:** 3 good
**Contribution:** 3 good
**Rating:** 7
**Confidence:** 3

**Summary:**

The authors study adaptive allocation of samples into control and treatment groups in an experiment. They (i) define new performance measures for such allocations, namely Neyman ratio and Neyman regret, (ii) introduce a new algorithm called Clip-OGD that achieves optimal Neyman regret, and (iii) provide asymptotically correct confidence intervals for experiments run with Clip-OGD.

**Strengths:**

Presentation-wise, this is an extremely well-written paper. In particular,

(i) The metrics that are proposed, Neyman ratio and Neyman regret, are well motivated. It is convincing why minimizing the Neyman ratio is desirable and how minimizing Neyman regret is a surrogate for that objective.

(ii) Clip-OGD is a simple and intuitive algorithm. Notably, it does not have hyper-parameters that require tuning (although they could potentially be tuned). This is a major strength for an online algorithm where cross-validation using an offline dataset would not be possible.

**Weaknesses:**

There are only minor weaknesses (please see my questions below).



**Questions:**

1) In the paragraph starting from line 226, Neyman regret is discussed from the perspective of multi-armed bandits. I understand how explore-exploit algorithms like UCB are not suitable for minimizing the Neyman regret; the purpose of adaptive Neyman allocation seems to be purely exploratory. Then, how does it relate to existing pure-exploration bandits?

2) Assumption 2 is not intuitive and is not explained well enough. What does "ruling out settings where the outcomes were chosen with knowledge of Clip-OGD" mean exactly? I understand the formulation assumes $y_0(t)$ and $y_1(t)$ are deterministic, but just to understand Assumption 2, what would its equivalent have been under a super-population assumption (i.e. if $Y_0(t)$ and $Y_1(t)$ were to be random and i.i.d.)?

3) Confidence intervals provided in Section 5 seem to be only asymptotically correct. Does this mean that no finite-sample guarantees are possible regarding the error rate of these intervals? This would limit the use of Clip-OGD in settings where strict error control is required (such as clinical trials).

4) Why do the experiments not include a multi-armed bandit baseline although they are mentioned as an alternative design in Section 6?

5) What does "informal" mean in Proposition 6.1? If it does not have a formal proof, maybe it should be stated as a remark rather than a proposition.

**Limitations:**

Yes, the authors adequately addressed the limitations of their work.

---

> ### Author Rebuttal · Authors · 2023-08-08
>
> # Response to Reviewer 4 (etbf)
>
> We thank you for your careful reading of our paper and we are happy to hear that you find the paper well-written and its results well motivated.
> Your comments in this review have been very helpful, as they have identified a few weak points in the paper which we have addressed.
>
> ## Question 1: Relation to Pure Exploration Bandits
>
> > In the paragraph starting from line 226, Neyman regret is discussed from the perspective of multi-armed bandits. I understand how explore-exploit algorithms like UCB are not suitable for minimizing the Neyman regret; the purpose of adaptive Neyman allocation seems to be purely exploratory. Then, how does it relate to existing pure-exploration bandits?
>
> Great question! Although there are many objectives for pure exploration bandits (e.g. simple regret, best arm identification), they are all concerned with selecting or identifying the best arm.
> In contrast, the objective in causal inference is to estimate the difference between the means of the two arms, to high precision.
> We are not only interested in *which arm* is the better arm, but rather we are interested in estimating *how much better* is the better arm.
> This is the average treatment effect.
>
> In Proposition 6.2, we showed that a bandit algorithm which minimizes cumulative regret will necessarily incur a large variance of the effect estimator, by orders of magnitude.
> This shows that cumulative regret and Neyman regret are generally incompatible.
> We have not investigated whether a similar incompatibility result holds for pure exploration bandits objectives such as simple regret and best arm identification, but we conjecture that the answer is "yes".
>
> ## Question 2: Assumption 2
>
> > Assumption 2 is not intuitive and is not explained well enough. What does "ruling out settings where the outcomes were chosen with knowledge of Clip-OGD" mean exactly? I understand the formulation assumes $y_0(t)$ and $y_1(t)$ are deterministic, but just to understand Assumption 2, what would its equivalent have been under a super-population assumption (i.e. if and were to be random and i.i.d.)?
>
> Thank you for your comment, which highlights the need for additional discussion following Assumption 2.
> Due to space considerations, we removed such a discussion.
> However, we are more than happy to add a brief discussion which addresses these points back in the main body.
> Please see a longer answer to your question in the "Global Response".
>
> ## Question 3: Confidence Intervals
>
> > Confidence intervals provided in Section 5 seem to be only asymptotically correct. Does this mean that no finite-sample guarantees are possible regarding the error rate of these intervals? This would limit the use of Clip-OGD in settings where strict error control is required (such as clinical trials).
>
> Yes, you are correct that we do not establish any finite sample guarantees for these confidence intervals.
> We agree that confidence intervals with finite sample guarantees would be more desirable in practice, provided that they would not increase dramatically in width.
>
> However, we remark that asymptotic coverage guarantees are the standard in causal inference (space is limited in this rebuttal, but we are happy to provide additional citations during discussion period).
> In fact, randomized clinical trials today typically use exactly these types of confidence intervals, which only guarantee coverage asymptotically.
> This is referred to in practice as a "large sample approximation".
> In this sense, our results are in-line with the literature.
> Moreover, over additional simulation results in the appendix show that the intervals cover at the nominal level.
>
> ## Question 4 -- MAB in Experiments
>
> > Why do the experiments not include a multi-armed bandit baseline although they are mentioned as an alternative design in Section 6?
>
> Thanks you for raising this point, which encouraged us to update our simulations and strengthened the paper.
> Please see "Global Response" for our discussion on the results of a MAB baseline and additional new simulation results.
> Initially, we did not include a MAB baseline because Proposition 6.2 shows that any adaptive design guarantees a cumulative regret of $\mathcal{O}(\sqrt{T})$ must incur a variance of $\Omega(1 / \sqrt{T})$, orders of magnitude larger than even naive Bernoulli.
> Simulations are consistent with this claim.
>
> ## Question 5: Proposition 6.1
>
> > What does "informal" mean in Proposition 6.1? If it does not have a formal proof, maybe it should be stated as a remark rather than a proposition.
>
> We recognize that "informal" was a confusing term to use in Proposition 6.1.
> Initially, the term "informal" was used here because the class of potential outcomes referenced by Proposition 6.1 is directly constructed in the appendix.
> Moreover, Proposition 6.1 is restated in the appendix with respect to this class.
> That being said, there is nothing "informal" about Proposition 6.1 from the point of mathematical rigor.
>
> To address this comment, we have removed the word "informal" and replaced the sentence after Proposition 6.1 with the following: "The specific class of potential outcomes referenced in Propposition 6.1 is constructed in Appendix E.1".

---

### Official Review · Reviewer_2JiC · 2023-07-08

**Soundness:** 3 good
**Presentation:** 4 excellent
**Contribution:** 3 good
**Rating:** 7
**Confidence:** 3

**Summary:**

The work considers adaptive experimental design for sequential experiments. To do so, a new regret-like measure, called Neyman regret is defined that compares the ratio of the variance under the chosen experiment design with respect to the variance under the optimal experiment design. Drawing connections to online convex optimization literature, an algorithm is developed that provides $\tilde O(\sqrt(T))$ Neyman regret. In contrast, it is also established that two-stage explore then commit or multi-arm bandit setups may not result in (super)linear Neyman regret. Additionally, asymptotically valid confidence intervals using the adaptively collected data are provided.

**Strengths:**

S1. An important topic of sequential experimental design, particularly setting up the framework from the point of regret minimization.

S2. The core idea is well-presented

S3. Comparison of Neyman regret with alternative designs is also valuable.

**Weaknesses:**

W1. Numerical simulations that analyzed more aspects of the algorithm could have made the paper stronger.

**Questions:**

A.  For the gradient term in the algorithm, it might be beneficial to elaborate on that, i.e., from variance eqn to eqn for estimating that variance and taking the derivative to get the cubic terms.

B. I see the need for the projection term in the algorithm, but it is not clear how to actually set it in practice.

C. I am curious if the authors can elaborate on the choice of the utility for gradient estimation. Instead of minimizing the utility with just the new sample, why not do a Follow-the-leader style algorithm to minimize the utility over all the samples seen so far? Maybe even FTRL, with a regularizer that prevents the distribution from shifting too far from Bernoulli(0.5) could replace the projection step?

D. I think the discussion around non-superefficient variance is important and I while I see the need of it, I do not quite understand it properly. Having more discussion about that could be beneficial.

E. What is the practical relevance of the experiment setup for Fig 1(b)?

**Limitations:**

F. To understand the sensitivity of the projection hyper-parameter, can ablations for it be provided in Fig 1?

---

> ### Author Rebuttal · Authors · 2023-08-08
>
> # Response to Reviewer 3 (2JiC)
>
> We thank you for your careful review of our paper and we are happy to hear that you believe the core idea to be well-presented and find the comparison to alternative designs valuable.
> We are grateful for the questions and concerns you raised in your review, which have strengthened the paper.
> We respond to your questions and concerns below.
>
> ## Weakness W1: Stronger Simulations
>
> > W1. Numerical simulations that analyzed more aspects of the algorithm could have made the paper stronger.
>
> Thank you for encouraging us to increase the strength of our simulations.
> We have added additional baselines and stability analyses.
> Please see "Global Response" for details.
>
> ## Question A: Gradient Term
>
> > A. For the gradient term in the algorithm, it might be beneficial to elaborate on that, i.e., from variance eqn to eqn for estimating that variance and taking the derivative to get the cubic terms.
>
> Thank you for highighting this weakness in our exposition.
> We will address this by adding a few sentences describing the derivation of the gradient estimator after Algorithm 1.
>
> ## Question B: Projection Term
>
> > B. I see the need for the projection term in the algorithm, but it is not clear how to actually set it in practice.
>
> In Clip-OGD, we set the projection parameter according to a decaying rate, $\delta_t = (1/2) \cdot t^{-1/\alpha}$, where $\alpha$ is a decay parameter chosen by the experimenter.
> Although $\alpha$ could be arbitrarily chosen, our regret analysis in the proof of Theorem 4.2 demonstrates that choosing $\alpha = \sqrt{5 \log(T)}$ minimizes our derived upper bound on the expected Neyman regret.
> In this sense, we recommend that experimenters choose $\alpha = \sqrt{5 \log(T)}$ in practice.
> Note that the experimental data is only collected once so that it is not possible to choose $\alpha$ using cross-validation or a similar statistical method.
>
> ## Question C: Gradient Estimation
>
> > C. I am curious if the authors can elaborate on the choice of the utility for gradient estimation. Instead of minimizing the utility with just the new sample, why not do a Follow-the-leader style algorithm to minimize the utility over all the samples seen so far? Maybe even FTRL, with a regularizer that prevents the distribution from shifting too far from Bernoulli(0.5) could replace the projection step?
>
> This is a great question!
> We tried an approach like this, but had difficulty analyzing its properties.
>
> On one hand, you need to allow the algorithm to be able to select $p_t$ arbitrarily close to $0$ or $1$ as $T$ grows, which means that penalizing the distance from Bernoulli(0.5) is challenging to implement correctly.
> On the other hand, an approach that does not explicitly constrain $P_t$ to be in the open interval $(0,1)$ might have the propery that the $P_t = 1$ or $P_t = 0$  at some round with non-zero probability, which would introduce bias in the Horvitz--Thompson estimator.
>
> An interesting open question is to achieve sublinear expected Neyman regret *without* using a projection step.
> We conjecture that this could shave off the sub-polynomial factor from our bounds on the expected Neyman regret and achieve a clean $\mathcal{O}(\sqrt{T})$.
>
> ## Question D: Assumption 2
>
> > D. I think the discussion around non-superefficient variance is important and I while I see the need of it, I do not quite understand it properly. Having more discussion about that could be beneficial.
>
> Thank you for your comment, which highlights the need for additional discussion following Assumption 2.
> Due to space considerations, we removed such a discussion.
> However, we are more than happy to add a brief discussion which addresses these points back in the main body.
> Please see "Global Response" for an interpretation of Assumption 2.
>
> ## Question E: Fig 1(b)
>
> > E. What is the practical relevance of the experiment setup for Fig 1(b)?
>
> Thank you for raising this question, which highlights a weakness in our exposition.
> We will update Section 7 with a brief discussion on the practical relevance of the experimental setup for Fig 1(b).
> Here is a longer discussion below:
>
> Practically speaking, Fig 1(b) represents a possible scenario where the first 100 people in the micro-economic experiment actually invest *less money* in machinery or equipment if they receive the macro-insurance intervention, while the remaining 14,435 people will invest *more money* given the macro-insurance.
> This sort of scenario is not unreasonable -- perhaps the first 100 people recruited into the trial were systematically different from the rest, i.e. from a well-connected part of the entrepreneurial community.
>
> Explore-then-Commit (ETC) experimental designs -- which are the predominant existing method proposed by (Hahn et al 2011) and (Blackwell et al 2022) -- heavily rely on the first units arriving being representative of later units.
> The simulation results in Fig 1(b) demonstrates that these methods can actually increase the variance, while Clip-OGD does not suffer from these issues.

---

### Official Review · Reviewer_NsFr · 2023-07-12

**Soundness:** 3 good
**Presentation:** 4 excellent
**Contribution:** 3 good
**Rating:** 4
**Confidence:** 3

**Summary:**

This paper studies the problem of “Adaptive Neyman Allocation”, which involves designing an efficient, adaptive experimental design. Neyman allocation is an infeasible experimental design which would be optimal (minimum variance) if the planner knew all the exact potential outcomes under different treatments. However, this is infeasible, and so the goal considered in the paper is to build an adaptive experimental design which is nearly as efficient (in terms of variance) as the infeasible non-adaptive Neyman allocation asymptotically.
To measure the performance, the first contribution of the paper is to propose new measures of regret (and regret ratio) similar to the notion of regret in bandits/statistical learning. Second, the paper proposes an adaptive design based on the idea of adaptive gradience descent to adjust the treatment probabilities to minimize regret. The paper shows that the regret in this approach scales as O(sqrt(T)). Finally the paper constructs confidence intervals which guarantee asymptotic coverage of the average treatment effect.


**Strengths:**

– Novelty: The paper claims to be the first to introduce the notion of Neyman regret in the context of adaptive experimental designs. I am not fully aware of the related literature, but this seems to be an interesting contribution to analyze from this perspective.

– Well-written: The paper is well organized overall and the concepts are introduced and explained crisply.

– Theory: The paper presents results well-grounded in theory.



**Weaknesses:**

1. Related Work: The related work section mainly talks about  Neyman allocation and about casual inference under adaptively collected data. However, it seems there is also a large body of work that studies adaptive experimentation/ adaptive design for randomized trials. All of these references seem to be missing (please see few examples below and references therein)? It would be good to distinguish this paper from this body of related work.

-- Eggenberger, Florian, and George Pólya. "Über die statistik verketteter vorgänge." ZAMM‐Journal of Applied Mathematics and Mechanics/Zeitschrift für Angewandte Mathematik und Mechanik 3, no. 4 (1923): 279-289

-- Xu, Yanxun, Lorenzo Trippa, Peter Müller, and Yuan Ji. "Subgroup-based adaptive (SUBA) designs for multi-arm biomarker trials." Statistics in Biosciences 8 (2016): 159-180.

-- Eisele, Jeffrey R. "The doubly adaptive biased coin design for sequential clinical trials." Journal of Statistical Planning and Inference 38, no. 2 (1994): 249-261.

-- Hu, Feifang, and William F. Rosenberger. "Optimality, variability, power: evaluating response-adaptive randomization procedures for treatment comparisons." Journal of the American Statistical Association 98, no. 463 (2003): 671-678.

2. Unsurprising: To me it is a little unsurprising/ unimpressive that the approach can achieve the optimal data efficiency as T tend to infinity (after a really large number of samples). The authors also seem to acknowledge this in part in the final section of the paper.

3. Empirical Evaluation: It may be useful to add more interesting baselines if available in comparison of the regret


**Questions:**

– In section 4.1 why isn’t the variance of adaptive experimental design, V also a function of t?

– There seem to be a few other approaches that study adaptive randomziation in clinical trials. For example: Zhang, Lanju, and William F. Rosenberger. "Response‐adaptive randomization for clinical trials with continuous outcomes." Biometrics 62.2 (2006): 562-569.
How does the proposed approach compare against existing methods?

– Ethical concern: Please see additional question under Limitations section.


**Limitations:**

Could there be fairness/social welfare concerns stemming from such optimal designs? Perhaps the allocation algorithm may assign a higher probability to a less effective or detrimental treatment simply because the variance in its outcomes is higher. As a result, in the quest for minimized variance, could the negative treatment be administered more often than advisable/necessary?

---

> ### Author Rebuttal · Authors · 2023-08-08
>
> # Response to Reviewer 2 (NsFr)
>
> We thank you for your thoughtful review of our paper and we are glad to hear that you found our exposition crisp and the results well-grounded in theory.
> We are grateful for the comments, questions, and concerns raised in your review: they have highlighted weak points in our paper, which we have worked to address.
> As a result, we believe the paper has been greatly strengthened.
>
> We are limited for space in our rebuttal, but happy to expand upon any points in the discussion period.
> We hope you find that many of these have been addressed.
>
> ## W1: Related Work
>
> > The related work section mainly talks about Neyman allocation and about casual inference under adaptively collected data. However, it seems there is also a large body of work that studies adaptive experimentation/ adaptive design for randomized trials. All of these references seem to be missing (please see few examples below and references therein)? It would be good to distinguish this paper from this body of related work.
>
> We thank you for providing us with additional references so that we can better contextualize our results within a broader literature.
> We have read these papers (some of them familiar, some of them not) as well certain references therein and prepared a (longer) appropriate survey which better distinguishes our paper from this body of work.
> To address this point raised in your review, we will include these papers into our literature review and better differentiate our work from the papers you raised.
>
> Our problem setting is distinct from this literature in two ways.
> The first distinction is the goal of the adaptive design.
> In the references above, there are a variety of goals from reducing sample size, to minimizing "harm", to identifying relevant subgroups.
> In contrast, our work focuses on the goal of minimizing the variance of the sequential Horvitz--Thompson estimator.
> The second distinction is the inferential frameworks.
> While the references you listed make strong independence and parametric assumptions (which simplify their problems), we take a design-based perspective, requirigin neither i.i.d. nor parametric assumptions on the outcomes.
>
> ## W2: Unsurprising
>
> > To me it is a little unsurprising/ unimpressive that the approach can achieve the optimal data efficiency as T tend to infinity (after a really large number of samples). The authors also seem to acknowledge this in part in the final section of the paper.
>
> We respectfully disagree; the state of prior work suggests that the main results presented in the paper are surprising and relevant to the design-based inference community.
> The most closely related work of (Hahn et al 2011) and (Blackwell et al 2022) for the same problem of Adaptive Neyman Allocation was not able to establish that the optimal Neyman variance was recovered in large samples -- in fact, we show in Proposition 6.1 that the adaptive designs presented in those papers *provably cannot* obtain the Neyman variance in large samples.
> Experimental results (Fig 1b) corroborate this theoretical analysis.
> Our main results are the first to provably achieve this goal of Adaptive Neyman Allocation.
> One of the insights of our work (which may be surprising to the design-based causal inference community) is that sophisticated algorithmic techniques seem to be required to achieve optimal Neyman variance, as the problem is equivalent to an adversarial online convex optimization problem.
>
> ## W3: Empirical Evaluation
>
> > It may be useful to add more interesting baselines if available in comparison of the regret.
>
> We have included two additional baselines.
> Please see "Global Response" for details.
>
> ## Q1: Variance Derivation
>
> > In section 4.1 why isn’t the variance of adaptive experimental design, V also a function of t?
>
> We believe there is a slight misunderstanding here -- you are absolutely correct that the variance of the adaptive experimental design $V$ will be a function of $T$.
> We write $V = (1 + \kappa_T ) \cdot V_N$, where $\kappa_T$ is the Neyman ratio and $V_N$ is the optimal Neyman variance.
> Both of these quantities depend on $T$.
> We are happy to further clarify if this is unclear.
>
> ## Q2: Comparison to Existing Method
>
> > There seem to be a few other approaches that study adaptive randomziation in clinical trials...How does the proposed approach compare against existing methods?
>
> Thank you for this additional reference.
> The recommended adaptive design in this paper is the Doubly Biased Coin Design (DBCD), which we have included in simulations (see "Global Response").
>
> ## Q3: Ethical Concerns
>
> > Could there be fairness/social welfare concerns stemming from such optimal designs? Perhaps the allocation algorithm may assign a higher probability to a less effective or detrimental treatment simply because the variance in its outcomes is higher. As a result, in the quest for minimized variance, could the negative treatment be administered more often than advisable/necessary?
>
> Great question!
> We have written an additional section for the appendix that addresses your point, which we are happy to share in the discussion phase.
>
> The highly influential 1979 "Bellman Report" writes that "even avoiding harm requires learning what is harmful...Learning what will in fact benefit may require exposing persons to risk".
> From this perspective, it may be ethically advisable to use a variance minimizing adaptive design because it allows the researcher to learn the effect while subjecting fewer human subjects to the experimental treatments.
> On the other hand, an adaptive treatment plan which minimizes cumulative regret will ensure that minimial harm is done to subjects in the experiment, but will offer less certainty about the extent of the benefit or harm of the treatments.
> Such an approach will lead to less informative generalizable knowledge of treatment effects, possibly defeating the goal of the research study and being, as a result, more unethical.

---

> > ### Comment · Area_Chair_b83D · 2023-08-21
> >
> > Hello Reviewer NsFr,
> >
> > Were your concerns regarding the novelty (surprise) addressed by the authors? I have reviewed the related works you suggested in the context of this paper, and while they are relevant, I do not believe their omission warrants rejection alone. Given the authors' efforts to bulk up the related work, are your concerns addressed?

---

### Official Review · Reviewer_xwuB · 2023-07-27

**Soundness:** 3 good
**Presentation:** 2 fair
**Contribution:** 2 fair
**Rating:** 5
**Confidence:** 1

**Summary:**

The paper proposes a new adaptative Neyman allocation for experimental design. The proposed adaptative design gets close to the optimal non-adaptative strategy without suffering from the same infeasibilities.

**Strengths:**

1) The writing and structure of the paper are OK.
2) The topic is very relevant, and the paper proposes adaptative variance with formal guarantees ensuring feasibility is interesting.

**Weaknesses:**

1) The paper is not easy to follow for the non-expert, both concerning the notations used and the derivations.
2) It seems that the motivation for this work could be broader. It is unclear to the reviewer why the authors focused on medical applications.
3) The method is only applied to a single microeconomic example.


**Questions:**

1) I would like to see some discussion about the medical applications vs a more broader motivation.

**Limitations:**

Limitations are adequately addressed.

---

> ### Author Rebuttal · Authors · 2023-08-08
>
> # Response to Reviewer 1 (xwuB)
>
> Thank you for your careful reading of our submission.
> We are glad to hear that you find the topic relevant and the formal guarantees interesting.
> Below, we respond to the specific critiques raised in your review.
>
> ## Broader Movtivations
>
> > It seems that the motivation for this work could be broader. It is unclear to the reviewer why the authors focused on medical applications...I would like to see some discussion about the medical applications vs a more broader motivation.
>
> Although clinical trials are a relevant application, we agree that the statistical methods in this paper apply more broadly to a variety of domains.
> Some motivating examples of randomized experiments in other disciplines include:
>
> - **International Development Economics**: International Development Economics (IDE) is concerned with determining which interventions (typically by NGOs or state governments) increase economic outcomes in developing nations. In the past 15 years, the introduction of randomized experiments (as opposed to relying solely on economic theory) has been a major development is figuring out "what works" (see, e.g. Banerjee et al 2015, Duflo 2012). In fact, Esther Duflo, Abhijit Banerjee, and Michael Kremer won the 2019 Nobel Prize in Economics for their foundational role in this line of work. IDE experiments with short-term outcomes can be implemented in a sequential manner.
> - **Email Marketing**: Nearly all companies that operate online use email marketing campaigns (see, e.g. Ellis-Chadwick & Doherty 2012). A common type of experiment is to test which personalization strategy will improve a certain outcome, e.g. a customer clicking a link. More generally, experiments like this are frequently run at tech companies to guide product development. These experiments typically have very quick response times and so can be made to run sequentially.
> - **Quantitative Political Sciences**: There has been a recent revolution in political science to incorporate more principled causal inference methods. A seminal work in this area is Gerber & Green (2000), which invesitgates the effects of canvassing on voter turn-out. More recently, Offer-Westort et al 2021 run a sequential experiment to test various interventions for reducing partisan bias.
>
> Our work shows that in settings where sequentially running the experiment is possible, fewer people are needed in the experiment to detect an effect, thereby saving money and resources.
>
> In order to address your comment, we will update the introduction to better highlight these applications beyond medical trials, thereby broadening the motivation for this work.
>
> - Banerjee, Abhijit, Esther Duflo, Rachel Glennerster, and Cynthia Kinnan. 2015. "The Miracle of Microfinance? Evidence from a Randomized Evaluation." American Economic Journal: Applied Economics, 7(1): 22-53.
> - Esther Duflo. “Women’s Empowerment and Economic Development”, Journal of Economic Literature, Vol. 50, No. 4: 1051-79, December 2012.
> - Ellis-Chadwick, Fiona and Doherty, Neil F. (2012). Web advertising: the role of email marketing. Journal of
> Business Research, 65(6) pp. 843–848.
> - Gerber, Alan S., and Donald P. Green. “The Effects of Canvassing, Telephone Calls, and Direct Mail on Voter Turnout: A Field Experiment.” The American Political Science Review, vol. 94, no. 3, 2000, pp. 653–63.
> - Molly Offer-Westort, Alexander Coppock, and Donald P. Green. "Adaptive experimental design: Prospects and applications in political science." American Journal of Political Science, 65(4): 826–844, 2021. doi: 10.1111/ajps.12597.
>
> ## Readability and Derivations
>
> > The paper is not easy to follow for the non-expert, both concerning the notations used and the derivations.
>
> Other reviewers (2JiC, etbf, wYBx) asked for further interpretation of Assumptions 1 and 2, as well as additional discussion into the derivation of the main algorithm, Clip-OGD.
> Please see our responses to their reviews for more details on how we addressed these points.
> We hope that these additional discussions on assumptions and derivations improve readability for a general audience.
>
> We agree that the notation used in design-based causal inference can be sometimes cumbersome, especially relative to general statistics or machine learning.
> We strive to make our paper as readable to as wide of an audience as possible, but we find value in sticking to conventional notation.

---

### Author Rebuttal · Authors · 2023-08-08

# Global Response

We thank the five reviewers for their careful reading of our paper.
We are happy to see that that the reviews were overall positive.
Moreover, we are very grateful for the critiques and questions from reviewers that revealed certain weaknesses in our submission.
We believe that the paper has been greatly strengthed as a result of this feedback.

We respond to each reviewer individually, but we list the main themes of the revisions here:

- **Simulations**: More simulations have been added, including a comparison to EXP3 and Doubly Biased Coin Design (DBCD) and an investigation into the stability of Clip-OGD relative to the step size.
- **Interpretation of Results**: We add additional discussions interpreting the assumptions and elaborating on the derivation of the algorithm.
- **Literature Review**: The literature review has been expanded and there are more careful comparisons to related prior work.
- **Ethical Considerations**: We have drafted a new section in the appendix that discusses ethical considerations grounded in The Bellmont Report.

We elaborate on some of these points below.

## Simulations

We thank the reviewers for encouraging us to strengthen our simluation results.
We have added two additional baselines for our simulations: EXP3 algorithm for outcome regret minimization and the Doubly Biased Coin Design (DBCD), as described in Section 6 of Eisele (1994).
See Figure 2 in the attached pdf for their performance.
We find that they suffer from variance which is orders of magnitude larger than what is achieved by Clip-OGD and ETC.
This is perhaps unsurprising because these adaptive allocation mechanisms are designed for different purposes.
The problem of (variance minimizing) Adaptive Neyman Allocation is relatively understudied so that we are unaware of additional meaningful baselines other than ETC.
We are happy to include these additional comparisons in the revised paper.

In addition, we have tested the stability of Clip-OGD to the step size, which can be found in Figure 1 of the attached pdf.
We tried various step sizes of the form $\eta = c / \sqrt{T}$ for $c \in \{ 0.25, 0.5, 1.0, 2.0, 4.0 \}$.
The original Clip-OGD was only run with $c = 1$.
We find that smaller step sizes improve convergence rates, effectively removing the "overhead of adaptivity" in this example.
However, because the randomized experiment can only be run once, experimenters will typically not be able to try many step sizes.
While it remains an open question about how to select a step size which best mitigates the "overhead of adaptivity", our recommendation of $\eta = 1 / \sqrt{T}$ still maintains good convergence properties.

## Interpretation: Assumption 2

Several reviewers asked for additional interpretation behind Assumption 2, which we provide below.

The Neyman regret is a quantity who expectation compares the variance of an adaptive design to the optimal Bernoulli design.
We show that when using Clip-OGD, we can bound the expected Neyman regret by $\mathbb{E}[\mathcal{R}_T] \leq \tilde{\mathcal{O}}(\sqrt{T})$.
However, the expected Neyman regret could be *negative* if the adaptive design achieves variance which is strictly smaller than the best Bernoulli design.
This seems unlikely to happen for "typical" outcomes, but it is not impossible.
In fact, an adversary could construct outcomes on which the optimal Bernoulli design achieves a variance like $V_N = \Omega(1/T)$, while *Clip-OGD specifically* will achieve a variance with a better rate, i.e. $V = \mathcal{O}(1/T^2)$.
In this case, $\mathbb{E}[\mathcal{R}_T] \approx - T$.
Note that this does not contradict the original regret bound that $\mathbb{E}[\mathcal{R}_T] \leq \tilde{\mathcal{O}}(\sqrt{T})$.
Such an adversary would have to have detailed knowledge of the adaptive design and so this construction of outcomes seems like a pathological edge case.
It is exactly this pathological edge case that Assumption 2 rules out.

Assumption 2 is very likely to hold automatically in an i.i.d. setting precisely because the potential outcomes are drawn by a distribution independently of treatment assignment.
In particular, the potential outcomes cannot be chosen by an adversary with knowledge of the experimental design.
We suspect that Assumption 2 would not be necessary in an i.i.d. setting, but proving this seems beyond the scope of the current paper.

---

### Decision · Program_Chairs · 2023-09-21

**Decision:**

Accept (spotlight)

**Comment:**

Under the potential outcomes model, the authors wish to estimate the average treatment effect of T trials (1/T) sum_{t=1}^T y_t(1) - y_t(0), where at each time t only y_t(Z_t) for Z_t=0 or Z_t=1 can be observed. By sampling Z_t ~ Bernoulli(p) one can obtain a consistent estimate as long as p in (0,1). While the most natural choice is p=1/2, some other value may have smaller variance based on the specific (a priori unknown) outcomes. If p* is this optimal mixing proportion, this paper considers an online setting where trials arrive sequentially t=1,2,… and the algorithm must decide some p_t in (0,1) to make the assignment. The objective is to compete with the optimal proportion p* in hindsight. While this problem definition is very simple and exceedingly natural, I myself (as well as the reviewers) are unaware of prior art that directly address this problem. Standard pure-exploration bandit algorithms, like most adaptive clinical trials, assume trials are IID and use adaptivity to maximize power. What is interesting is that this work drops this IID assumption. While I agree with some reviewers that it is not so surprising that one can compete with the best static allocation in hindsight, it is not at all obvious how one might achieve it. And moreover, there is great value in new problem statements that provide a new perspective on an old problem setting. The reviewers raised a number of questions and concerns, but nothing that was not appropriately addressed by the authors or warranted second-guessing.